



# 1 Potential impact of aerosols on convective clouds revealed by
# 2 Himawari-8 observations over different terrain types in
# 3 eastern China

Tianmeng Chen[a,b,c], Zhanqing Li[b,*], Ralph A. Kahn[b,d], Chuanfeng Zhao[a,*], Daniel
Rosenfeld[e], Jianping Guo[c], Wenchao Han[a,b], Dandan Chen[c]
a. State Key Laboratory of Remote Sensing Sciences, and College of Global Change
and Earth System Science, Beijing Normal University, Beijing, 100875, China
b. Department of Atmospheric and Oceanic Sciences & Earth System Science Interdi
sciplinary Center, University of Maryland, College Park, Maryland 20740, USA
c. State Key Laboratory of Severe Weather, Chinese Academy of Meteorological
Sciences, Beijing 100081, China
d. Earth Sciences Division, NASA Goddard Space Flight Center, Greenbelt, Maryland
20771 USA
e. Institute of Earth Sciences, Hebrew University, Jerusalem, Israel
* To whom all correspondence should be addressed
Zhanqing Li
Email: zli@atmos.umd.edu
Chuanfeng Zhao
Email: czhao@bnu.edu.cn





**Abstract**
Convective clouds are common and play a major role in Earth's water cycle and energy
balance; they may even develop into storms and cause severe rainfall events. To
understand the convective cloud development process, this study investigates the
impact of aerosols on convective clouds by considering the influence of both
topography and diurnal variation of radiation.    By combining texture analysis,
clustering and thresholding methods, we identify all convective clouds in two warm
seasons (May-September, 2016-2017) in eastern China based on Himawari-8 Level 1
data. Having a large diurnally resolved cloud data together with surface meteorological
and environmental measurements, we investigate convective cloud properties and their
variation, stratified by elevation and diurnal change. We then analyze the potential
impact of aerosol on convective clouds under different meteorological conditions and
topographies. In general, convective clouds tend to occur preferentially under polluted
conditions in the morning, which reverses in the afternoon. Convective cloud fraction
first increase then decrease with aerosol loading, which may contribute to this
phenomenon. Topography and diurnal meteorological variations may affect the strength
of aerosol microphysical and radiative effects. Updraft is always stronger along the
windward slopes of mountains and plateaus, especially in northern China. The
prevailing southerly wind near the foothills of mountains and plateaus is likely to
contribute to this windward strengthening of updraft and to bring more pollutant into
the mountains, thereby strengthening the microphysical effect, invigorating convective
clouds. By comparison, over plain, aerosol decreases surface heating and suppresses
convection by blocking solar radiation reaching the surface.
**Key Words**
Himawari-8; convective cloud; aerosol-cloud interactions, topographic effects on cloud



## 1. Introduction

Convective cloud is important for Earth's energy balance and the water cycle.
Changes in the distribution or triggering time of convective cloud can have a large
impact on the climate system (Stocker et al., 2013). Previous studies have shown that
aerosol particles in the atmosphere can affect the formation and development of
convective clouds, through both radiative and microphysical effects (Ramanathan et al.,
2001; Tao et al., 2012; Altaratz et al., 2014; Rosenfeld et al., 2014a, 2014b; Li et al.,
2016; Zhao et al., 2018a; Yang et al., 2019), which can dramatically affect the weather
and climate(Zhao et al., 2020).
Aerosol particles generally cool the surface by scattering the downward solar
radiation (McCormick and Ludwig, 1967; Charlson et al., 1992), whereas light-
absorbing aerosol can additionally warm the atmosphere above the surface (Ackerman
et al., 2000). These aerosol radiative effects change the surface radiation budget and the
atmospheric temperature profile directly, thereby altering the onset time of convective
cloud formation and precipitation (Feingold et al., 2005; Wang et al., 2013; Zhou et al.,

2020).

Aerosols, by acting as cloud condensation nuclei (CCN) and/or ice nuclei (IN), can
also affect the formation and growth of cloud droplets (Rosenfeld, 2000; Kaufman et
al., 2002; Garrett et al., 2004; Lohmann and Feichter, 2005; Zhao et al., 2012), which
is regarded as the aerosol microphysical effect. For constant cloud liquid water content,
increases in aerosol particle number concentration can produce more but smaller cloud
droplets, thus increasing cloud albedo (Twomey and Warner, 1967; Twomey, 1974).
This effect, called the "Twomey effect", is a well-established influence on cloud
properties in polluted environments (Kaufman and Nakajima, 1993; Feingold et al.,
2003). Smaller cloud droplets can increase cloud lifetime and reduce precipitation
(Albrecht, 1989), whereas for some deep convective cloud, the latent heat released by
the formation of more and smaller ice particles can invigorate deep convection and





increase rain rate (Andreae et al., 2004; Kaufman et al., 2005; Rosenfeld et al., 2008;
Li et al., 2011; Fan et al., 2013; Li et al., 2019). For cases of thin clouds, smaller cloud
droplets enhance cloud thermal emissivity, trap more longwave radiation within the
atmosphere and alter cloud development (Garrett and Zhao, 2006; Zhao and Garrett,

2015).

Many previous studies show that the interactions between aerosols and weather

variables are complex (Stevens and Feingold, 2009; Altaratz et al., 2014), making it
challenging to untangle aerosol effects from meteorological factors that influence
convective clouds. As different types of clouds form under different meteorological
conditions, some previous observational studies attempted to classify clouds into
different types, to distinguish aerosol effects in various meteorological regimes
(Andreae et al., 2004; Khain et al., 2005; Li et al., 2011; Gryspeerdt and Stier, 2012;
Gryspeerdt et al., 2014; Qiu et al., 2017). Other studies consider different cloud types
as different stages of convective development, to identify aerosol effects on convective
cloud evolution, stratified by meteorological conditions (Chen et al., 2016; Guo et al.,

2018).

Meteorological conditions can change during the day and can vary from one place to

another. For example, convective cloud and precipitation maxima tend to occur in the
early morning over open ocean, but in the late afternoon or early evening over land,
driven primarily by temporal differences in the radiative forcing (Chang et al., 1995;
Garreaud and Wallace, 1997; Sui et al., 1997; Zhou et al., 2008; Li et al., 2010). Other
studies show that the impact of terrain on the convective cloud can also be complex
(Roe, 2005; Houze Jr., 2012), and that the meteorological factors that control
convection can be affected by terrain (Romatschke and Houze Jr., 2010, 2011a, b).
Diurnal variation of solar radiation can alter wind circulation in a valley, and therefore
may control the spatial distribution of convective cloud in mountainous regions
(Kirshbaum and Durran, 2004, 2005; Kirshbaum et al., 2007; Romatschke et al., 2010).



In recent years, studies also attempted to explore the effects of topography on
aerosol-precipitation interactions. Lynn et al. (2007) simulated the aerosol effect on
orographic precipitation using the WRF model and found that the intensity of
orographic precipitation is suppressed in more polluted environments. However, when
the cold rain process is involved, precipitation is delayed but intensified  (Givati and
Rosenfeld, 2004; Rosenfeld and Givati, 2006; Xiao et al., 2015; Yang et al., 2016c).
When light-absorbing aerosol is present, radiative heating in the atmosphere can
produce enhanced instability, which may trigger disastrous precipitation, especially late
in the day (Fan et al., 2015). These studies revealed some potential influences of aerosol
on orographic precipitation from either model-simulations or from several specific
observed cases. Only a few studies include sufficient observational data to analyze the
relationships between aerosol and orographic convective cloud statistically. This might
be due to a lack of observational data in the past with sufficiently high spatial and
temporal resolution.
Although polar orbiting satellites provide high spatial resolution data, they cannot
track the diurnal development of convective clouds. In contrast, geostationary satellite
data can track the evolution of convective clouds over the entire day at a slightly lower
spatial resolution (Chakraborty et al., 2015, 2016). The launch of the Japanese next-
generation geostationary satellite Himawari-8 in October 2014 rendered an opportunity
to study aerosol effects on convective clouds throughout the day. The diurnal variation
in different terrain types may be particularly strong due to amplified variations in
radiation and sunshine duration by terrain (Li and Weng, 1987, 1988, 1989). The current
study aims to characterize the differences between convective clouds in polluted and
clean environments, untangling the probable influence of topography on the way
aerosols affect convective clouds, and to explore how the effects of aerosols change
diurnally. To achieve this goal, we first develop an automatic method of identifying
convective clouds using geostationary satellite data, and then assess how aerosol effects
change during the day and in different topographic regions of eastern China.





131 The paper is organized as follows: Section 2 introduces the study region and data

132 selection. The method is described in Section 3. Section 4 presents the temporal and

133 spatial distributions of convective cloud and discusses the possible impact by aerosol

134 on convective cloud over different terrains. In section 5, we give a brief summary.

135 ## 2. Study region and data

136 ### 2.1 Region of interest

137 Rapid economic and industrial growth has brought heavy pollution to China,

138 especially to eastern China, in recent decades. Thus, high aerosol loading over this area

139 provides a natural laboratory for us to study the aerosol impact on convective clouds

140 (Guo et al., 2011). In addition, eastern China has relatively complex topography, where

141 convective clouds can be triggered and develop differently, leading to convective cloud

142 regimes having distinctive diurnal change patterns. We show the terrain distribution and

143 the mean concentration of particles with aerodynamic diameters smaller than 2.5 μm

144 ($PM_{2.5}$) during May-September in 2016-2017 over eastern China in Figure 1. Generally,

145 terrain height (TH) tends to increase from east to west in this region, and $PM_{2.5}$ mass

146 concentration is generally higher over the plains and lower over mountain ranges and

147 plateaus. In order to investigate how these landscapes impact cloud fraction and

148 convective cloud diurnal variation, we chose the area with longitudes from 105°E to

149 125°E, and latitudes from 20°N to 45°N as the region of interest (ROI) for this study.

150 ### 2.2 Data

151 a. Himawari-8 data

152 The first in a new generation of Japanese geostationary meteorology satellite, named

153 Himawari-8, was launched in 2014. It carries a state-of-art optical sensor, the Advanced

154 Himawari Imager (AHI), which can provide significantly high resolution observations

155 of the Earth system from space (0.5 or 1 km for visible and near-infrared bands at the

156 sub-spacecraft point and 2 km for infrared bands) and time (around 10 min for Full Disk

157 and 2.5 min for sectored regions) from space (Bessho et al., 2016). These advantages



make it possible to detect rapid weather changes, especially the triggering and
development of convective cloud. The geostationary sub-spacecraft point is located at
140.7°E over the equator, so most of our ROI is covered.
In this study, we use the Himawari-8 L1 gridded data from the Japan Aerospace
Exploration Agency (JAXA) P-Tree system to develop a convective cloud
identification method and investigate how aerosols impact convective clouds over
eastern China. This dataset is generated by the Earth Observation Research Center
(JAXA/EORC) from Himawari Standard Data, with re-sampling to equal latitude-
longitude grids. (For more detail, see https://www.eorc.jaxa.jp/ptree/userguide.html ;
last accessed: Oct. 2018 .) The channels we use here are centered at 0.64 μm, 11.2 μm,
and 12.4 μm, with a spatial resolution of 0.02°×0.02° and a temporal resolution of 10
min.
b.  MODIS cloud mask
We use the MODIS/Aqua MYD35 cloud mask data (Wilson et al., 2014) to validate
the convective cloud identification method developed here. Cloud mask data at 1 km
resolution     (from     the     MYD35     Cloud_Mask     product;     web     address:
https://modis.gsfc.nasa.gov/data/dataprod/mod35.php; last accessed: Sep. 2018) is used
to compare with the near-simultaneous cloud identification result from our method.
c.  Particulate matter (PM) data
In previous studies, aerosol optical depth (AOD) (Andreae, 2009;Niu and Li,
2012;Wang et al., 2018), visibility (Chen et al., 2016), aerosol concentration with
diameters between 100 nm and 3 μm (Zhao et al., 2018b;Yang et al., 2019;Zhao et al.,
2019), and particulate matter up to 10 μm in diameter ($PM_{10}$) (Guo et al., 2016) were
used as proxies for cloud concentration nuclei (CCN). However, satellite AOD
retrievals can only be made in cloud-free conditions, and near-cloud retrievals are
frequently influenced by cloud contamination (Li et al., 2009). In addition, remote-
sensing methods cannot distinguish the part of the CCN size spectrum smaller than





about 0.05 μm from atmospheric gas molecules. On the other hand, particulate matter
can be measured from the surface or aircraft under all-sky conditions. Particle size up
to 10 μm may be much larger than the typical scale of CCN, so we consider particle
matter up to 1 μm ($PM_1$) or 2.5 μm ($PM_{2.5}$) in size as a CCN proxy. Due to the limited
availability of $PM_1$ measurement in eastern China, we chose $PM_{2.5}$ as an indicator of
different CCN levels in the environment for this study.
Figure 1 also shows the mean value of $PM_{2.5}$ measured at 1205 ground-based stations
across eastern China during the warm months (May-September) in 2016-2017. The
average $PM_{2.5}$ mass concentration generally lies between 20 and 60 μg/m$^3$, with higher
values over the Beijing-Tianjin-Hebei region. Although not uniformly distributed, the
ground stations cover almost all regions in eastern China. Note that these stations
provide hourly observations of $PM_{10}$ and $PM_{2.5}$ concentrations.
d.   MERRA-2 reanalysis
In order to assess the impact of meteorological factors on convective clouds, and to
analyze the dependence of aerosol effects on convective cloud fraction, we adopt
meteorological variables from the second Modern-Era Retrospective analysis for
Research and Applications (MERRA-2) reanalysis dataset (web address:
https://gmao.gsfc.nasa.gov/reanalysis/MERRA-2/; last accessed: Oct. 2019). MERRA-
2 is the latest atmospheric reanalysis produced by NASA's Goddard Earth Observing
System Data Assimilation System Version 5 (GEOS-5), using a new generation of
satellite observation sources from 1979 to the present (Gelaro et al., 2017). In this study,
we adopt the temperature at 2 m, relative humidity at 700 hPa and 850 hPa, vertical
velocity at 925 hPa and 850 hPa, and surface specific humidity to evaluate how
convective cloud fraction changes with respect to terrain height under different
meteorological conditions, and to gain insight into whether the aerosol effects are
independent of other factors that might influence convective cloud triggering and
development during daytime. We obtained all these parameters at a spatial resolution



of 0.5°×0.5°, and a temporal resolution of 3 hours. Based on the findings in previous
studies, we chose the following factors to characterize the dynamics and
thermodynamics of the environment:
*Lower-tropospheric stability (LTS)*. The lower-tropospheric static stability is defined
by Klein and Hartmann (1993) as the inversion strength of the atmosphere. Chen et al.
(2016) adopted this LTS definition to distinguish between stable and unstable cloud
profiles and to analyze the independence of aerosol effect on clouds. In this study, we
use this parameter to assess how the lower atmosphere stability influences the
performance of convective clouds. The formula of LTS can be written as

$$ LTS = \theta_{700\ hPa} - \theta_{surface} \tag{1} $$


$\theta_{surface}$, which represent the potential temperature at the surface, is calculated from the
air temperature at 2 m; the potential temperature at 700 hPa ($\theta_{700\ hPa}$) is directly
extracted from MERRA-2 pressure level dataset.
*Potential Temperature*. Temperature, especially lower-level atmospheric temperature,
plays a critical role in triggering the development of convective clouds. As air
temperature decreases systematically with altitude in most places, it is not proper to
directly compare temperatures over different terrain. As the potential temperature is
conserved regardless of height, it reflects the near-surface heating to some extent.
Additionally, Wang et al. (2018) point out that selecting potential temperature avoids
some duplication of temperature and humidity information. Thus, potential temperature
is adopted in this study.
*Vertical velocity*. Vertical velocities at 850 hPa ($\omega_{850}$) and 925 hPa ($\omega_{925}$) are chosen
to investigate the role of vertical airflow in convection. As vertical motion over different
terrain may vary, this factor can produce large differences in convective cloud
occurrence frequency. We use $\omega_{850}$ and $\omega_{925}$ to represent the low-level dynamical
conditions for terrain above and below 1000 m, respectively. $\omega > 0$ represents downward
air motion, whereas $\omega < 0$ means the air motion is upward. Uncertainty lies in the





different distances between the 850 hPa and 925 hPa level and cloud base. There are
likely large differences in the updraft strength between these two levels and the cloud
base in some cases. However, in the formation of convective clouds, the vertical
velocity between the surface and cloud base is more essential in transporting vapor and
energy to higher levels (Lee et al., 2019). As these two levels cover most of the surfaces
above and below 1000 m and can reflect the vertical velocities beneath cloud to some
extent, we chose to use them as reference values to represent the dynamical conditions
when convective clouds occur.
*Relative humidity*. Water vapor supply is essential to the formation and development
of convective clouds (Redelsperger et al., 2002;Chakraborty et al., 2018) and is a crucial
component of cloud water condensation and evaporation in aerosol-cloud interactions
(Altaratz et al., 2014). As convective cloud formation is more sensitive to the under-
cloud and near-surface water vapor content, similar to vertical velocity, relative
humidity (RH) data at 700 hPa ($RH_{700}$, for regions with TH$\geqslant$1000 m), 850 hPa ($RH_{850}$,
for regions with TH<1000 m) and the specific humidity at the surface (q) are employed.

## 3. Methods

### 3.1 A convective cloud identification method

Numerous previous studies have attempted to develop methods for detecting and
classifying cloud features, including convective clouds, based on satellite observations.
One of the most common methods for identifying cloud is thresholding (Williams and
Houze Jr., 1987). However, Wielicki and Welch (1986) found that the identified cloud
fraction depends strongly on threshold values, as cumulus cloud reflectance varies
greatly within individual clouds and at cloud edges. As a result, many studies now adopt
digital image processing to help identify shallow convective clouds more accurately.
One widely used image processing method is textural analysis, which adopts second-



order statistics representing the texture of the digital image, as first proposed by
Haralick (Haralick et al., 1973). This method, known as the gray level co-occurrence
matrix (GLCM) method, was applied by Welch et al. (1988a, b) in their LandSat data
analysis of marine stratocumulus cloud texture. The GLCM is a matrix of
counts/frequencies of grey values for pairs of pixels, whose relative positions are
defined by the polar coordinates (d, θ).   The formula can be written as:
$GLCM(i,j,d,\theta) = Pr\{I(x_1,y_1) = i, I(x_2,y_2) = j\} \; x_1, x_2 \in m \; and \; y_1, y_2 \in n$   (2)
where Pr{E} denotes the probability of event E, $I$ represents the image matrix of size
m×n, $(x_1,y_1)$ and $(x_2, y_2)$ are the two elements in $I$ with gray tone value i and j, which
are separated by distance d in direction θ. The unit of d is a pixel, and θ always takes
0°, 45°, 90°, and 135°. The maximum difference between i and j defines the size of
GLCM. When the frequency of the $(i, j)^{th}$ element is more concentrated near the off-
diagonal of the GLCM, the image contains more complex texture. To define texture
properties from the GLCM, several image statistical variables are derived from this
matrix, including "contrast", "homogeneity", "energy", and "entropy" (Haralick,
1979;Welch et al., 1988b;Baum et al., 1997;Bottino and Ceballos, 2014). The "contrast"
measures the intensity contrast between a pixel and its neighbors, assessed over the
entire image. "Homogeneity" measures the closeness to the diagonal of the GLCM
element distribution. "Energy", also termed the angular second moment, measures the
complexity of the image, and "entropy" measures the degree of randomness, evaluated
over the entire image. The formulas are:
$$Contrast = \sum_{i,j} |i-j|^2 GLCM(i,j) \qquad (3)$$
$$Homogeneity = \sum_{i,j} \frac{GLCM(i,j)}{1+|i-j|} \qquad (4)$$
$$Energy = \sum_{i,j} GLCM(i,j)^2 \qquad (5)$$
$$Entropy = \sum_{i,j} GLCM(i,j) \ln[GLCM(i,j)] \qquad (6)$$



The objective of this section is to identify new-born and mature convective clouds.
As the edges of convective clouds tend to be very sharp (Purdom, 1976), large
differences between i and j can produce large 'contrast' values (Equation 3). Thus, in
this study, we use the mean "contrast" data at d=1 in the four directions ($\theta$=0°, 45°, 90°,
and 135°) to identify convective clouds.
Besides this parameter, we also employ the visible reflectance (VIS, 0.64 $\mu$m) and
brightness temperatures ($T_b$) at 11.2 $\mu$m and 12.4 $\mu$m to help identify the distinctive
patterns of convective clouds. The spatial gradient of $T_b$ (11.2 $\mu$m) helps exclude very
low-elevation fogs, whose temperatures are close to that of the surface in the morning.
We use these three parameters in a k-means clustering analysis. Those clusters with
relatively higher "contrast" (with mean Contrast>3.5) are considered either small
convective clouds or the edges of mature convective clouds.
Unlike stratus clouds produced by large-scale systems, mature local convective cloud
tops tend to have very high VIS reflectance and small area (Lima and Wilson, 2008).
As the cloud tops of mature convective clouds are also relatively flat, they produce
small "contrast" values. We consider those clusters having area smaller than 40,000
km$^2$ (10,000 pixels), mean VIS reflectance larger than 0.75, and maximum VIS
reflectance larger than 0.9 as the tops of mature convective clouds.
In addition, we adopt the split window technique to exclude cirrus (Mecikalski and
Bedka, 2006). The brightness temperature differences between 11.2 $\mu$m and 12.4 $\mu$m
are near-zero for convective clouds, and we exclude those pixels with brightness
temperature differences numerically smaller than -4K. This allows us to produce a
cloud mask with a high probability of isolating convective cloud. Figure 2 shows the
entire flowchart of our convective-cloud identification method.
As we identify convective clouds using the combined results from texture analysis,
clustering, and thresholding, we name this cloud identification method the Texture-
Clustering-Thresholding-Convection IDentification (TCT-CID) method.





**3.2 Validation of the convective cloud mask**

To validate the TCT-CID method, we compare the convective clouds identified with our cloud mask against the MODIS/Aqua MYD35 cloud mask. As the MODIS product does not classify clouds into different types, we use a scene from the hilly regions in southern China at 13:40 LT on July 30[th], 2016 as an example. It contains a vast convective cloud field, and most of the clouds in this scene are convective clouds. Figure 3a shows the true-color image of the scene we chose. Figures 3b and 3c are the MODIS cloud mask data and the convective cloud mask from our method, respectively. We can see that there is a good agreement between the MODIS cloud mask and that identified by our method. As we have screened the cirrus with the split-window method to isolate only convective clouds, the cloud area identified by our method is smaller than the cloudy area found by the MODIS cloud mask. Nevertheless, the majority of convective clouds are well captured by our method.

In order to validate the result of convective clouds identified by the TCT-CID method statistically, we compared the identified convective cloud mask with the Himawari-8 Level 2 cloud type data from the L2CLP010 product (see https://www.eorc.jaxa.jp/ptree/userguide.html; last accessed: May 2020). This product provides the cloud type information using the ISCCP cloud classification criteria. The frequencies of different cloud types corresponding to the identified convective cloud masks are shown in Figure S1. We can find that the frequencies of DCC and Sc are relatively significant, especially around noon time, which indicates that the TCT-CID method is effective at identifying deep convective clouds and stratocumulus clouds. Other cloud types, such as altostratus, nimbostratus and cumulus cloud also show relatively large amounts. These cloud types can be seen as representing different stages in the development of convective clouds. Although their frequencies only exceed the median but not the 2σ values of the distributions, they still show significant differences from the frequency of cirrus, stratocirrus, altocumulus and stratus. After 16:00, as the solar zenith angle grows, the cloud top reflectance increases significantly. The criteria



of the TCT-CID method become less strict, so that cloud identification errors increase.
Nevertheless, the frequency of DCC still passes the 2σ line, which implies that deep
convection is the most robust cloud type that can be identified by the TCT-CID method.

These results suggest that the identification by the TCT-CID method is relatively

reliable in studying the convective cloud properties and their relationship with aerosols.
**4. Results and discussion**
**4.1 The diurnal cycle of convective clouds**

We use the L1 data from Himawari-8 acquired from May to September in 2016 and

2017 to identify local convective clouds over eastern China and build the spatial
distribution of convective clouds. Figure 4 shows the frequency of convective clouds
occurring between 08:00 and 17:00 LT, aggregated over the entire study period. This
includes 20455 Himawari images, containing more than 2000 samples within each pixel
within each hour. Convective cloud occurs predominantly over the sea in the morning,
and gradually shifts inland after local noon. This convective cloud occurrence pattern
is driven by distinct differences in the specific heat capacities and boundary-layer
thermodynamics over land and ocean. In the morning, radiative cooling and land-breeze
contribute to the formation of convective clouds over the ocean, whereas temperatures
near the land surface tend to be relatively low, which makes it difficult for convective
clouds to form there at this time of the day. But in the afternoon local time, as land
surface temperatures increase, the near-surface air layer becomes more unstable,
making it easier for convective clouds to form. The accumulation of land-surface
heating during the day also favors the development of deep convection in the afternoon.
Such patterns are highly consistent with previous studies (Garreaud and Wallace,
1997;Sui et al., 1997;Li et al., 2010).

To further validate the identification results, the statistical patterns of convective

cloud masks are investigated in Figure 4 and 5. The impact of diurnal solar radiation
variation and topography on convective clouds are already well-understood (Houze Jr.,




1993;Houze Jr., 2012;2014), and thus, these results serve as further support for our
cloud identification method. The gray lines in Figure 4 mark the 1000 m surface
elevation contour, which is approximately the boundary between the plains and elevated
terrains. Around 13:00-14:00 local time, convective clouds begin to form over the
elevated regions, and the amount begins to increase afterwards (Figure 4).
Figure 5 shows the joint distribution of convective cloud occurrence frequency (CC
OF) with respect to terrain height (TH) over land and associated meteorological factors.
Vertical velocities (relative humidity) at 850 hPa and 925 hPa (700 hPa and 850 hPa)
are used as proxies for the basic state of dynamics and water vapor for regions with
elevation above and below 1000 m, respectively.
Generally, CC OF has two peaks, at heights below 500 m and above 1000 m (Figure
5a-d), which is likely due to the different sample sizes over terrain of different heights
(Figure S2). Nevertheless, the thermal factors that impact convective cloud occurrence
show strong differences between lower and higher elevation surfaces. Convective
clouds tend to require more unstable thermal conditions over regions with higher
surface elevation compared to regions with lower surface elevation (TH≤1000 m). The
LTS can be 5~10K lower, whereas the potential temperature $\theta$ can be nearly 10K higher,
for convective clouds to be favored at higher elevations. Further, instability and surface
heating are also stronger when convective cloud occurrence is favored after about 11:00
LT relative to the early morning. For dynamical conditions, shown in Figure 5i-p, the
mean values of CC OF for both $\omega_{850}$ and $\omega_{925}$ are negative (i.e., upward motion) and
tend to move slightly in the negative direction and become stronger from 08:00-14:00,
acting as a nonnegligible contributor to the diurnal variation of convective clouds
during the day.
The under-cloud relative humidity (RH) values ($RH_{700}$ and $RH_{850}$ for regions of
TH>1000 m and TH≤1000 m, respectively) show a general increase from 08:00 to
14:00 and are close to 100% when cloud occurrence is most common. In addition, the



under-cloud RH over lower terrain is always higher than it is over higher-altitude
regions during daytime (Figure 5q-x). This pattern may be caused by the stronger
surface heating over higher terrain, which decreases the RH near the surface. Also note
that where convective clouds occur, specific humidity at the surface (q) is always higher
for regions having TH≤1000 m than for regions where TH>1000 m (Figure 5y-z2). This
might indicate that, compared with higher terrain regions, it is always moister over
lower terrain, so it is more difficult for cloud droplets to evaporate in such places. q
over regions with TH>1000 m decreases more rapidly than for lower altitude regions
from 11:00 to 17:00, but $RH_{700}$ increases at higher altitudes, probably indicating that
moisture over higher terrain is transported to higher altitudes to form convective clouds
at these times.
All these thermal, dynamical and moisture conditions show relatively significant
differences between lower and higher terrains in our data, which may reflect distinct
impacts of topography on the formation and development of convective clouds. Such
patterns show that our data conform to the expected patterns supports their use for the
analysis presented below.
**4.2 Changes in convective cloud diurnal cycle associated with aerosol**
In this study, $PM_{2.5}$ observations from 1205 stations over eastern China are used as
proxies for CCN, to roughly separate clouds into polluted and clean classes. We match
the hourly measured $PM_{2.5}$ with the 10 min convective cloud identification results by
supposing the convective cloud observed in the same hour occurs under the same $PM_{2.5}$
conditions. For each site, we use the top quarter of the $PM_{2.5}$ concentration distribution
as the criterion for identifying polluted cases, and the bottom quarter as the clean cases
(Figure 6). We then aggregate them into a 0.4°×0.4° grid using nearest neighbor
interpolation. Only those clouds with centroids located within a grid box classified as
polluted (clean) are deemed as polluted (clean) convective clouds. We calculated the
convective cloud fraction (CCF) using the number of convective clouds under polluted
(or clean) conditions divided by the total convective cloud amount within each grid cell.





On average, $4.6\times10^7$ ($4.1\times10^7$) pixels are deemed as clean (polluted) convective cloud
within each hour. Figure 7 shows the difference of CCF in polluted and clean
environments. Warm (cold) colors in the figure mean that there are more (less)
convective clouds under the polluted condition. Additionally, only those data points that
exceed the 95% significance level according to the Pearson's $\chi^2$ test are plotted. From
Figure 7a to 7e, we find that before 12:00 LT, convective clouds under polluted
conditions generally occur in larger amounts, especially over the plateau region and
some of the mountain regions. This pattern reverses gradually from 13:00 to 17:00 LT
(Figure 7f-7j): the amount of convective clouds under polluted conditions tends to
diminish, relative to those in clean conditions, in the afternoon. However, in several
places, the CCF difference generally persists from morning to afternoon. Some red dots
in southern and eastern China seem to occur over megacities, presumably caused by the
co-action of high aerosol loading and the urban heat island effect (e.g., for the
megacities around Yangtze River Delta (YRD) and the Pearl River Delta (PRD),
marked by black circles in the figure). Furthermore, complex topography may also lead
to different convective cloud response to aerosol loading. Over northern China, more
convective clouds form over mountains under polluted conditions, whereas over the
central China plain, with mountains around, convective clouds may be suppressed all
the time by high aerosol loading. The number distribution of convective cloud clusters
in each area bin, aggregated over the entire ROI is shown in Figure S3. Polluted
convective cloud covers a larger area than clean convective cloud early in the day; this
pattern gradually reverses after 13:00 LT, starting from the decrease in number of
smaller convective cloud clusters. And after 14:00 LT, convective cloud area under
clean conditions dominates. This pattern may suggest that high aerosol loading is
probably one of the factors inhibiting the formation of small convective clouds via the
aerosol radiative effect in the afternoon.
In an attempt to assess the effects of pollution on diurnal convective cloud behavior,
the influence of other meteorological factors is addressed later in section 4.4 by





stratifying the data based on such factors. We note here only that a relatively clear
diurnal pattern in CCF exists, and that pollution effects appear to be correlated with this
pattern.

Koren et al. (2008) demonstrated two opposite effects of absorbing aerosol on cloud

cover, i.e. the microphysical effect and the radiation effect, by theoretical derivation,
and verified this theory with observations in the Amazon region. Their study concluded
that aerosol particles can increase cloud droplet number by serving as CCN. However,
when aerosol concentration is higher, the attenuation of solar radiation by aerosol
particles decreases the surface temperature, and atmospheric heating inhibits moisture
flux, thus suppressing convection. As aerosol loading increases, surface temperature
tends to decrease regardless of aerosol type (Gu et al., 2006;Jiang et al., 2013;Yang et
al., 2016a;Yang et al., 2016b;Yang et al., 2018). Thus, we can infer that under conditions
of high aerosol loading, the vertical moisture flux may be suppressed, which would
inhibit convective cloud formation. Therefore, investigating the diurnal variation of
aerosol microphysical and radiative effects might help explain the patterns shown in
Figure 7.

Figure 8 shows the relationship between convective cloud and $PM_{2.5}$ concentration.

Ten equally sampled bins of $PM_{2.5}$ concentration were defined, and we calculated the
mean CCF within each bin using the convective cloud amount in each $PM_{2.5}$ bin divided
by the total convective cloud amount within the same area for all $PM_{2.5}$ values. Sample
sizes are shown in the color shaded background in each subfigure, and the mean sample
number within each $PM_{2.5}$ bin is $\sim 8 \times 10^5$. The three-point moving average of the values
is also plotted. We find that CCF first increases with respect to $PM_{2.5}$ mass concentration
and then starts to decrease; this pattern persists throughout the day. The $PM_{2.5}$ mass
concentrations at all turning points of the curves are between 20 $\mu g/m^3$ and 30 $\mu g/m^3$.
Similar results were also found by previous studies (Guo et al., 2017;Jiang et al.,
2018;Wang et al., 2018). Adding to these previous results, we find that the relationship



of $PM_{2.5}$ and CCF persists throughout the day, as we have used high-resolution
geostationary satellite data that provide us ample samples at different times.
Furthermore, the aerosol effect on convective cloud is probably robust not only for deep
convective clouds, but also for convective clouds at any stage of development. This
pattern might be attributed to the competition between the microphysical effect
dominating at low $PM_{2.5}$ concentration, and the radiative effect becoming increasingly
important at higher $PM_{2.5}$ concentration.
The CCF values corresponding to the average thresholds identifying clean and
polluted conditions (marked as vertical red line pairs with numbers in each Figure 8
panel) generally tend to be higher under polluted than under clean conditions from
08:00-11:00 LT, whereas from 12:00-17:00 LT the pattern gradually reverses. The
shapes of the moving average curves change slightly from morning to afternoon
compared with the mean CCF over all times (magenta dots), where CCF is lower in the
morning before the tipping point but higher in the afternoon, and after the tipping point,
CCF is higher in the morning but lower in the afternoon. As we found from Figures 5
and 7, meteorological condition changes associated with topography and diurnal solar
radiation variations may play roles in altering the shapes of the CCF curves. For
conditions with $PM_{2.5}$ concentrations lower than the 20~30 μg/m$^3$ turning zone, more
convective cloud is formed, probably due to more unstable environments and stronger
surface heating, especially in the afternoon. But as the surface is generally moister
before noon time (Figure 5x-z2), especially at higher terrains, the higher q may suppress
cloud droplet evaporation and thus keep CCF from sharply decreasing.
**4.3 Effects of topography on the aerosol-convective-cloud relationship**
In order to isolate the probable effect of topography on the aerosol-convective-cloud
relationship, we further investigate in this section the CCF changes along with TH at
different levels of aerosol loading. The mean CCFs at different TH in both polluted and
clean conditions are shown in Figure 9. The CCF under clean (or polluted) conditions
is calculated using the formula shown below:





$$\text{CCF}_{C(P)}(i,j;h,t) = \frac{N_{C(P)}(i,j;h,t)}{N_{total}(i,j;h)} \times 100\% \qquad (7)$$
where $N_{C(P)}(i,j;h,t)$ represents the number of convective clouds occurring under clean
(C) or polluted (P) conditions in the $(i,j)^{th}$ pixel box in ROI in elevation bin $h$ during
hour $t$, and $N_{total}(i,j;h)$ represents the total number of convective clouds observed in
each pixel box at elevation bin $h$ during the daytime. Sample sizes are shown in Figure
S2. We find that the CCF difference between polluted and clean conditions generally
agrees with Figure 7 in that CCF is higher for polluted cases in the morning, lower in
the afternoon, and the differences are statistically greatest in early morning and late
afternoon. In addition, the CCF differences between polluted and clean conditions vary
considerably along with increasing TH, which may indicate that the effects of
topography and air quality on CCF co-vary, and the impact of topography might be
much stronger compared with increased aerosol loading.
There is also another aspect of these phenomena. In Figure 10, by normalizing the
occurrence frequencies by the total number of polluted and clean cases within each hour,
respectively, we explore how topography changes the polluted and clean convective
clouds spatially. The formula for this normalized CCF (NCCF) can be written:
$$\text{NCCF}_{C(P)}(i,j;h,t) = \frac{N_{C(P)}(i,j;h,t)}{\sum_i \sum_j N_{C(P)}(i,j;h,t)} \times 100\% \qquad (8).$$
The mean NCCF is calculated within each elevation bin $h$. Unlike CCF (Equation 7),
the denominator for NCCF (Equation 8) is not summed over all times-of-day; because
the elevation-related response reverses over the day (e.g., Figure 9), NCCF focuses
more specifically on how CCF at a given location and terrain elevation compares with
all locations at the same elevation and the same time, reducing the influence of diurnal
variation and emphasizing elevation-related differences. As such, the difference in
NCCF between clean and polluted cases reflects the difference caused by topography
when the overall environment is under clean or polluted conditions. We can see from
Figure 11 that below the elevation of 500 m, most of the convective clouds are

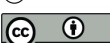



suppressed under polluted conditions, whereas over regions with terrain height greater
than 1000 m, especially before 14:00 LT, the amount of convective cloud under polluted
conditions is significantly larger. This phenomenon may partly explain the results
shown in Figure 7, where complex topography plays an important role in the aerosol
effect on convective clouds. Under polluted conditions, convective clouds over lower
terrain are much easier to suppress, whereas over elevated terrain, convective clouds
are more likely to be invigorated.
As the topography over eastern China has a general step-like distribution, we
roughly separate the terrain heights into four bins representing the plains (0-500 m),
mountain ranges (500-1000 m), plateau (1000-1500 m) and high mountains (1500-2000
m), and assess how the aerosol effect changes over different topography. The pixel
number within each bin is over $4.0\times10^5, 1.6\times10^5. 2.3\times10^5$ and $0.5\times10^5$. We calculate
the CCF within each of the 10 equally sampled $PM_{2.5}$ bins over each sub-region. Figure
11 shows how cloud fraction changes with respect to $PM_{2.5}$ concentration over the four
different terrain elevation ranges. In all four sub-regions, CCF first increases with $PM_{2.5}$
concentration and then decreases. But before the turning zone (between 20~30 μg/m³)
the CCF is slightly higher over mountains and plateaus than over the plains, but after
the turning zone, this pattern reverses. This probably occurs because instability and
surface heating are stronger over the higher terrain, which invigorates the convective
cloud development by enhancing the aerosol microphysical effect (Rosenfeld and
Lohmann, 2008). The air over the plain regions is moister than that over the mountain
and plateau regions, which suppresses droplet evaporation thus tends to overtake the
aerosol radiative effect.
In order to further investigate the probable effect of topography in aerosol-
convective-cloud relationship, the vertical circulation and moisture distributions are
studied. The meridional-vertical distribution of relative humidity and wind are shown
in Figure 12. Generally, the circulation pattern is consistent throughout the day, the



updraft south to 35°N is always stronger than that over the northern region, and the
relative humidity in this region is significantly higher. Under clean conditions, the
strong updraft and southerly wind south to 30°N may bring large amount of moisture
inland, contributing to convection development in this region. But under polluted
conditions, the southerly wind is weaker, and the relative humidity above this region is
lower, which may lead to a restraint of convective cloud development due to the
inhibition of aerosol microphysical effect. In regions north to 35°N, there is an obvious
north wind component under clean conditions, bringing dry and clean air to this region,
which may suppress the development of convective clouds. But under polluted
conditions, under the control of a relatively strong southerly wind, moisture and
pollutants may be blown toward the mountain regions and forced to be uplifted by the
elevated terrain, strengthening the formation and development of convective clouds,
especially over the windward slopes north to 40°N. The zonal-vertical wind and relative
humidity changes are shown in Figure S4 and S5. The updraft along windward slopes
is always stronger, especially under polluted conditions in the northern part of the ROI.
This is likely to contribute to strengthening the aerosol microphysical effect over such
regions, which facilitates the invigoration of convective clouds. All the patterns
described above agree well with the phenomena identified in the previous figures,
indicating that different circulation patterns and the changes associated with different
topography may have a considerable impact on the variability of the aerosol-
convective-cloud relationship.
**4.4 The environmental dependence of the aerosol effect**

To further isolate the signal of aerosol effects from that of meteorological conditions,

and to characterize the co-variation of topography and aerosol effects on convective
clouds, the changes of CCF with aerosol loading under various thermodynamic,
dynamical and humidity conditions at different time-of-day and over different terrain
heights are shown in Figures 13 and 14. Note that vertical velocities (relative humidity)
at 850 hPa and 925 hPa (700 hPa and 850 hPa) are used to represent the basic dynamics



(water vapor) state for regions with elevation above and below 1000 m, respectively.
We defined ten equally sampled bins of $PM_{2.5}$ concentration and calculated the joint
distribution of CCF along with each $PM_{2.5}$ and meteorological factor bin, the sample
sizes in each subfigure is shown in Table S1 and S2. Generally, the CCF along with
$PM_{2.5}$ concentration shows unimodal distributions, CCFs increase at first and then
decrease with $PM_{2.5}$ concentration, as we saw in Figure 8, but now in each of the vertical
velocity and relative humidity strata, i.e., regardless of differences in meteorological
conditions. The patterns are consistent at different time-of-day and over different
topography, which may indicate that under different thermodynamic, dynamical and
water vapor conditions the competition between the aerosol microphysical and
radiation effects always exists, no matter how meteorological conditions vary, at least
within the study domain.
The mean tipping point of CCF curves at different values of meteorological variables
is marked as the turning line between the increasing and decreasing CCF trends (dashed
lines in Figures 13 and 14). From Figure 13a-h, the turning line moves to smaller $PM_{2.5}$
values as surface heating increases and the atmosphere becomes more unstable from
morning to late afternoon. The peaks of CCF move from stable (weak surface heating)
conditions to unstable (strong surface heating) conditions during the day (Figure 13a-h
and 14a-h). This phenomenon may prove that changes in thermodynamic conditions
attributed to diurnal solar radiation variation and topography are probably among the
impact factors that influence the changing CCF curve shapes in Figures 8 and 11.
Stronger surface heating and more unstable conditions may increase CCF when the
microphysical effect of aerosol dominates the CCF changes, and would probably
strengthen the aerosol microphysical effect.
Updraft changes, which can represent under-cloud dynamical conditions, at 850 hPa
and 925 hPa for terrain higher and lower than 1000 m, respectively, are relatively small
during the day and over different topography. However, the turning values of CCF
generally decrease and CCF peaks occur under stronger updraft conditions as well



(Figure 13i-p). This pattern may indicate that more aerosol particles are entrained into
the clouds from the boundary layer when uplift is stronger, which in turn might
strengthen the aerosol microphysical effect. Further, as shown in Figure 14k-l, stronger
updrafts over higher mountains (with TH>1500 m) invigorate convective clouds after
the turning line, especially for $PM_{2.5}$ concentrations higher than 50 $\mu g/m^3$, which
suggests that the suppression of convective clouds by aerosol radiative effect is
counteracted.
For water vapor conditions, both higher relative humidity below cloud base (Figures
13q-x and 14q-x) and higher specific humidity at surface (Figures 13y-z2 and 14y-z2)
generally produce larger CCF. Higher RH and higher q are also associated with higher
CCF peaks. These patterns indicate that moister conditions can lead to greater activation
of aerosol particles, which may strengthen the aerosol microphysical effect, and might
overtake the suppression from the aerosol radiative effect in higher aerosol loading
conditions in these regions.
The meteorological factors that influence the aerosol effect on convective clouds are
very likely to interact with one another, which may produce combined impacts on
convective clouds and lead to nonlinear changes to the CCF distribution, creating large
variations in the results. By analyzing the probable co-variation of aerosol effects,
meteorological factors and the impact of topography on convective clouds, we find that
the CCF changes caused by both the aerosol microphysical and radiative effects are
robust under a range of meteorological conditions, whereas the strength of these two
effects can be influenced by specific thermodynamic, dynamical and humidity
conditions.
However, testing whether the results are due mainly to aerosol effects is only a first
step; establishing proof of the mechanisms by which aerosol affects convective cloud
occurrence is another important question. If synoptic factors, differences in
meteorological conditions associated with topography and aerosols work together,



determining which factors dominate the formation and development of convective
clouds needs to be explored with deeper mining of the data, as well as modeling studies,
in the future.

## 650 5. Summary

Following rapid economic development and industrialization, eastern China has
faced increasingly severe air pollution during recent decades. Aerosols, which play an
important role in the formation and development of clouds and precipitation, can be
among the main factors influencing urban inundation, hail and severe storms.
The interaction between aerosol, weather, and topography is complex, so untangling
their effects, that jointly influence convective cloud formation, is difficult. This study
applies very large, diurnally resolved geostationary satellite data and extensive ground-
based observations to investigate the characteristics of convective clouds, the impacts
of aerosol on convective cloud properties, and the potential mechanisms that define the
aerosol impacts on convective clouds under different meteorological conditions and
over different topography. Having such large data sets allows us to stratify by various
factors and isolate patterns among the multiple dimensions. The key results of this study
are as follows:
• We develop a convective cloud identification process named the TCT-CID
algorithm by combining the merits of texture analysis, a clustering technique, and
a threshold method, using the Level 1 data from the Japanese geostationary satellite
Himawari-8 during the period from May to September in 2016 and 2017. The
method offers stable and relatively accurate performance in identifying convective
clouds.
• The cloud mask is used to study the occurrence frequency and the regional
distribution of convective clouds over eastern China, first to determine whether the





new data reproduce expected cloud-occurrence patterns. Statistical results show that convective cloud occurrence frequency (CC OF) is higher under more unstable conditions with stronger surface heating and updraft. And in the afternoon, this phenomenon is more significant than in the morning. The increases in both under-cloud relative humidity and surface specific humidity produce higher CC OF during the day. There is also a significant difference between higher and lower terrain regions. The consistency of these patterns with previous studies and the classic theories of convective cloud formation helps validate the results of the TCT-CID algorithm.

- We then compared convective clouds under clean and polluted conditions, and further examined the possible impact of the diurnal cycle and topography on the aerosol-convective-cloud relationship. We find that the convective cloud fraction generally tends to be larger before noon and smaller in the afternoon under more polluted conditions, but megacities and complex topography can influence the pattern. This result provides a new perspective on the aerosol effect compared to previous studies.

- A relationship between aerosol loading and convective cloud fraction is found. The cloud fraction increases initially, but then decreases with successive increases in aerosol loading. This pattern is likely due to the combined action of the aerosol microphysical and radiative effects. Previous studies found similar results for deep convective cloud and convective precipitation studies, but by using high-resolution geostationary satellite observations, we further find that this pattern is probably robust for convective clouds at all stages of development.

- Although the aerosol-convective-cloud relationship is relatively stable, some variability also exists. The pattern varies throughout the day depending on terrain height, and is modulated by varying thermodynamic, dynamical and humidity conditions during the day. We find that the meteorological variations driven by



diurnal solar radiation changes and topography are probably among the reasons for
changes in the relative strength of aerosol microphysical and radiative effects. Over
higher terrains such as mountains and plateaus, a southerly wind component is
likely to contribute to the strengthening of the microphysical effect through forced
uplifting of pollutant and moisture, which invigorates convective cloud over the
windward slopes, whereas over plains areas, aerosol pollution blocking solar
radiation is likely to dominate, thus decreasing the surface heating and suppressing
convection by the enhancing the radiative effect. However, as aerosol concentration,
synoptic meteorological factors and topography might work together in influencing
the formation and development of convective cloud, the phenomena found above
can also be affected by nonlinear interactions among these factors.
Moreover, the analysis of this study is based mainly on satellite observations, which
in themselves provide limited insight into the mechanism underlying the observed
patterns. In the current study, we aimed only to isolate the possible effects of aerosol
on convective cloud properties under different meteorological conditions. However,
further exploration, including model simulations and/or targeted in-situ or aircraft
observations, are still needed to reveal the specific mechanisms behind these
phenomena.

## 718     Acknowledgment

We acknowledge the JAXA P-tree system (http://www.eorc.jaxa.jp/ptree/) for
providing the Himawari-8 Level 1 data, and the Global Modeling and Assimilation
Office (GMAO) (https://gmao.gsfc.nasa.gov/reanalysis/MERRA-2/), for providing
MERRA-2 reanalysis data. We would also like to extend our sincerest thanks and
appreciation to the China National Environmental Monitoring Center for providing the
hourly PM$_{2.5}$ data. This study is supported by the Natural Science Foundation of China
(41925022, 91837204), the Ministry of Science and Technology of China National Key



R&D Program on Monitoring, Early Warning and Prevention of Major Natural
Disasters (2017YFC1501403).

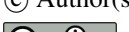



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



**Figures**

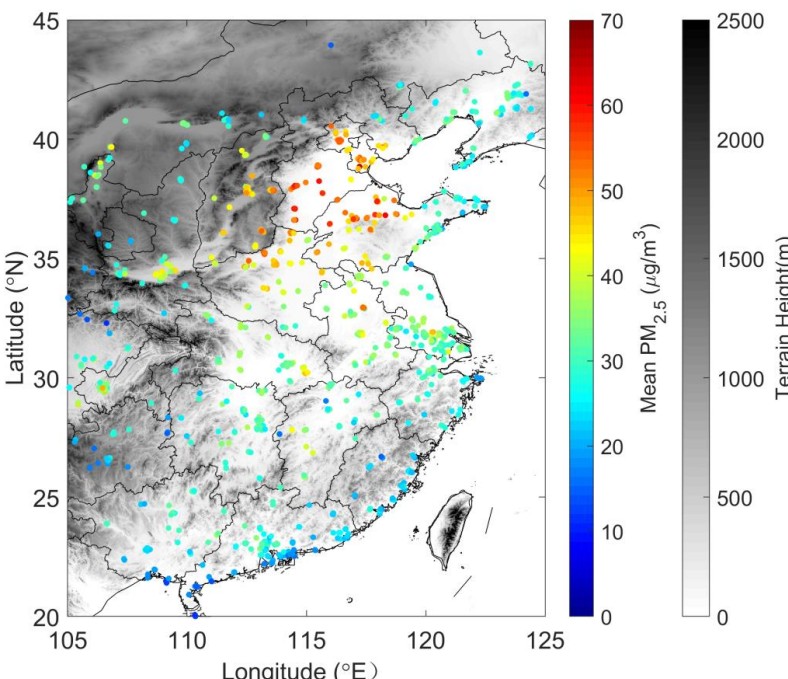


**Figure 1**. Surface elevation and the mean value of ground-based PM$_{2.5}$
measurements over eastern China. The terrain height (TH) of this region is
represented with gray shading. Colored dots show the mean PM$_{2.5}$ concentration
from 1205 surface stations during May-September in 2016-2017 over this area.





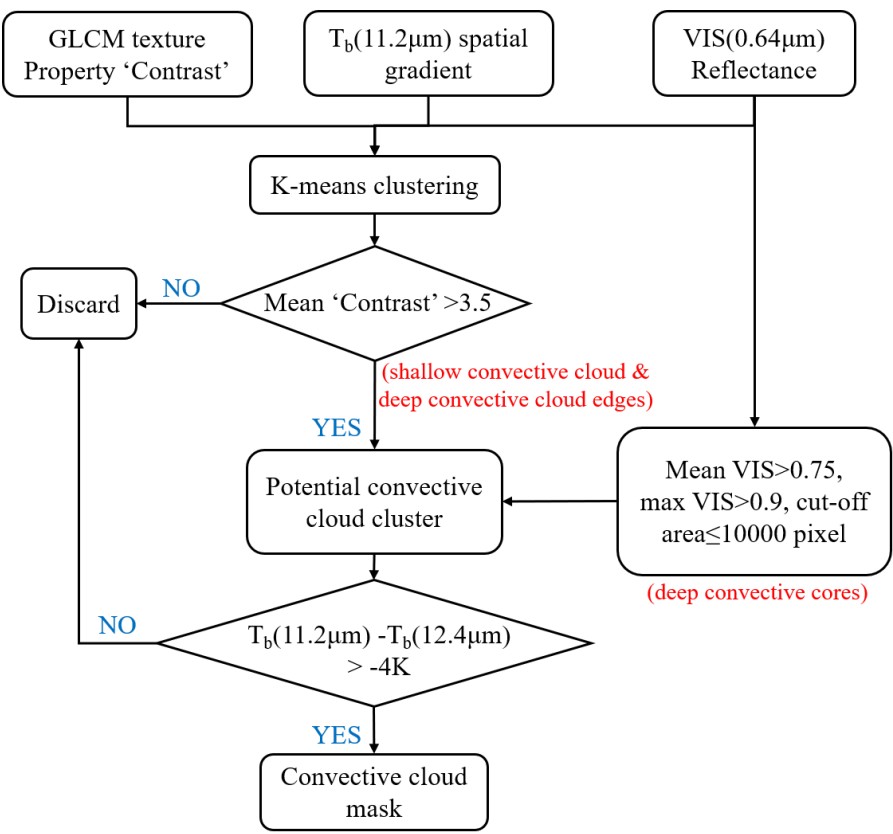


**Figure 2**. Flowchart of the convective cloud identification method.



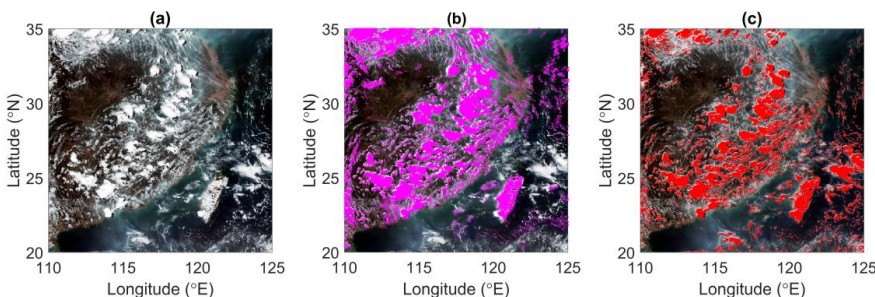


**Figure 3.** Comparison between MODIS cloud mask and the cloud identified by our convective cloud (CC) mask, at 13:40 LT on July 30[th], 2016. (a) The true color image, (b) MODIS cloud mask data from the MYD35 product (magenta points), and (c) convective clouds (CC) identified using the TCT-CID method.





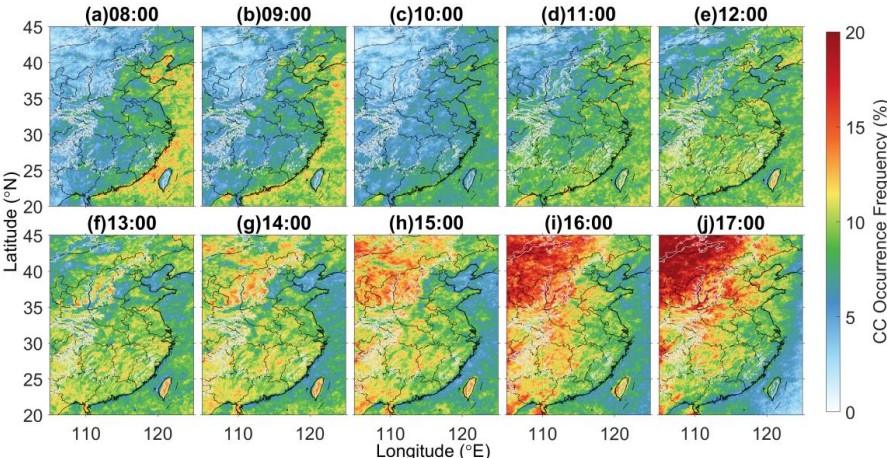


**Figure 4**. Diurnal cycle of convective cloud (CC) occurrence frequency (OF) during
daytime. The gray contour lines in each figure represent the terrain height (TH) at 1000
m. Most locations west of the contour line have TH >1000 m, whereas terrain east of
the contour lines has TH <1000 m.

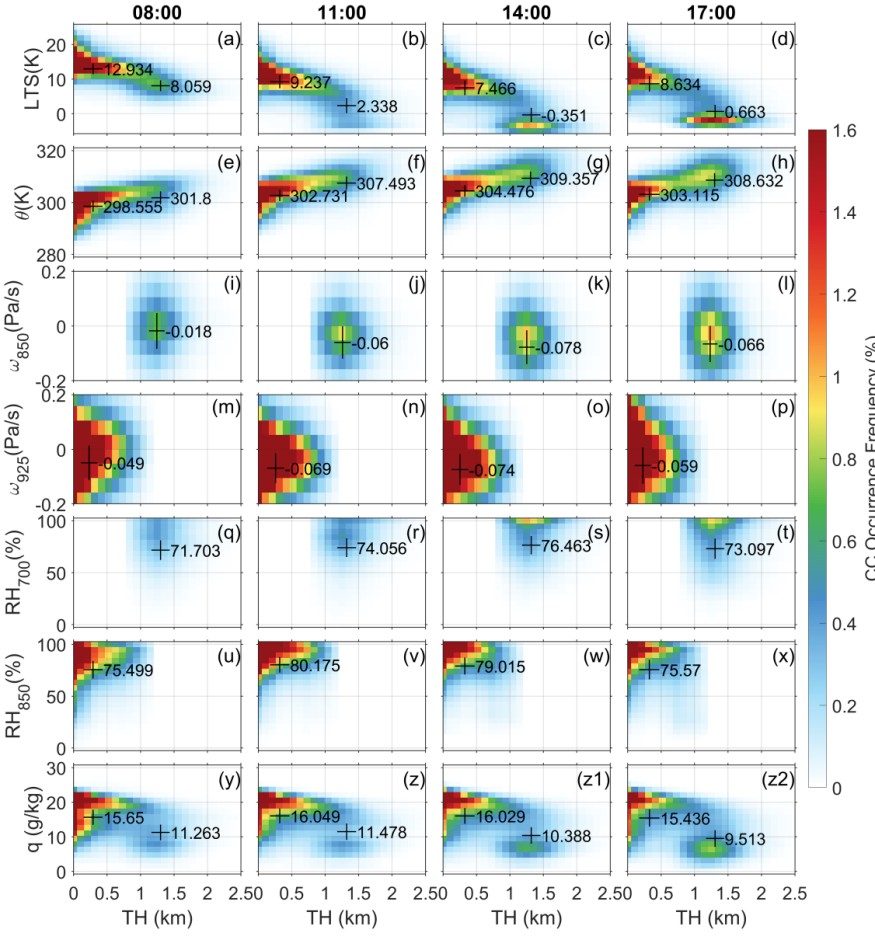


**Figure 5**. Over-land convective cloud (CC) occurrence frequency (OF) with respect to terrain height (TH) and (a-d) lower-tropospheric stability (LTS), (e-h) potential temperature ($\theta$), (i-l) vertical pressure velocity at 850 hPa ($\omega_{850}$) over terrain above 1000 m, (m-p) vertical pressure velocity at 925 hPa ($\omega_{925}$) over terrain below 1000 m, (q-t) relative humidity at 700 hPa ($RH_{700}$) over terrain above 1000 m, (u-x) relative humidity at 850 hPa ($RH_{850}$) over terrain below 1000 m, (y-z2) specific humidity at surface (q). The columns represent local time over the ROI of 08:00, 11:00, 14:00 and 17:00, respectively. Black crosses and numbers beside mark the mean values of the variables over regions with terrain height below 1000 m and above 1000 m, crosses mark the



standard deviations of the variables.

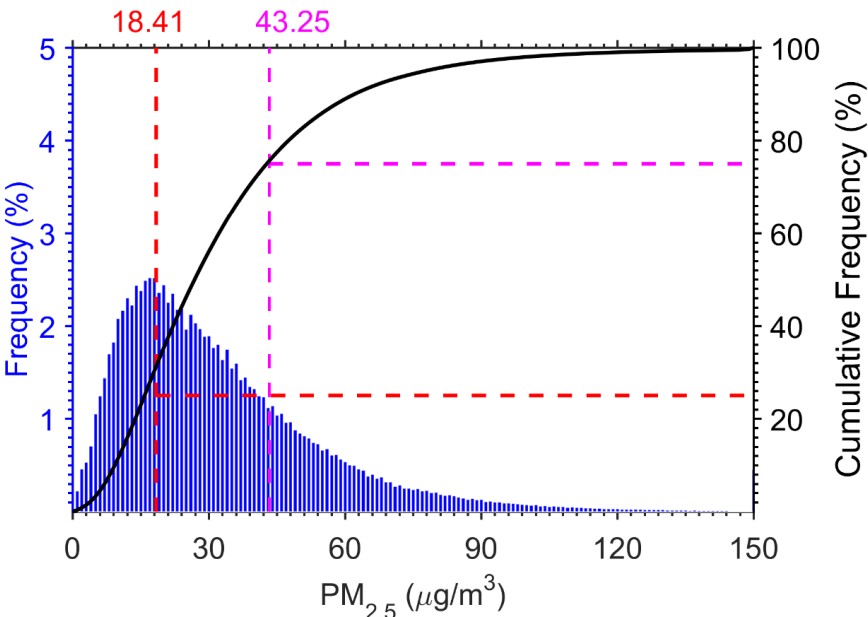


**Figure 6**. Mean PM$_{2.5}$ concentration distribution for 1205 sites over Eastern China during daytime (08:00-17:00 local time). Blue bars and the black line show the frequency and cumulative frequency of mean PM$_{2.5}$, respectively. The mean value of the top quarter is marked by the magenta dashed line, and the mean value of the bottom quarter is marked with the red dashed line.








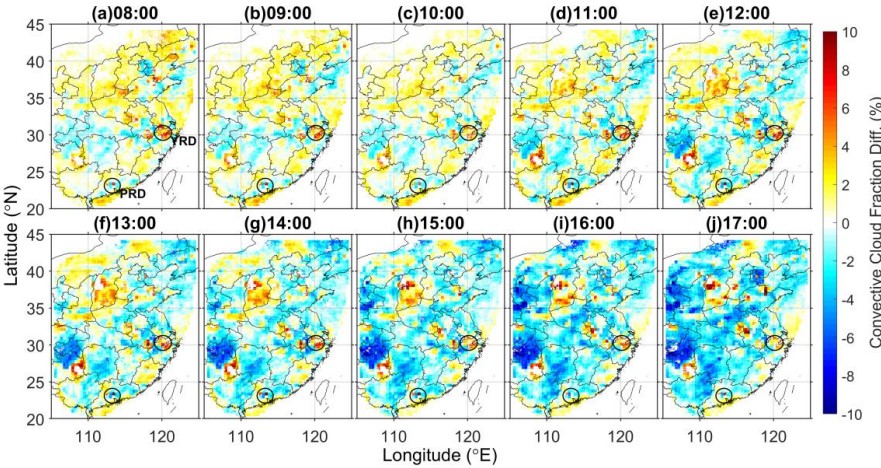

**Figure 7**. Diurnal changes of convective cloud fraction (CCF) difference between

polluted and clean conditions (Polluted-Clean) during May-September in 2016-2017.

Time marked above each figure is the local time. Black circles mark the Yangtze River

Delta (YRD) and Pearl River Delta (PRD). (Note that grid points are plotted only if

they exceed the 95% significance level ($p < 0.05$) according to the Pearson's $\chi^2$ test).





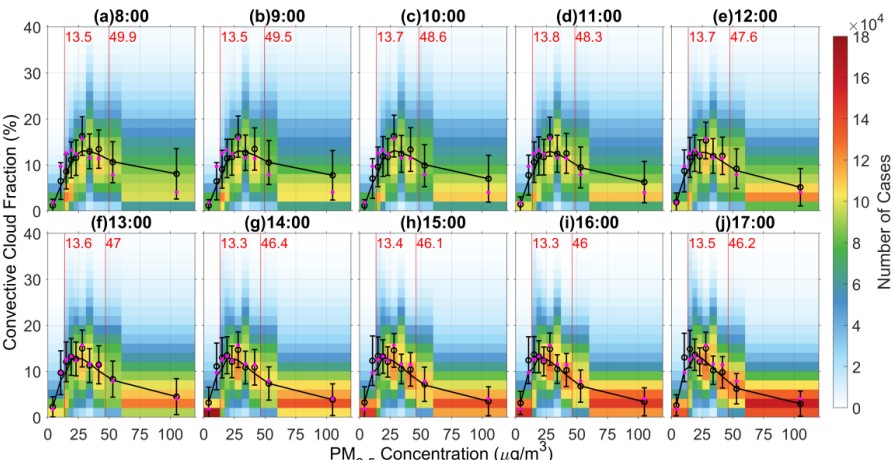

Figure 8. Convective cloud fraction with respect to $PM_{2.5}$ concentration at different times of day during May-September, 2016-2017. Color shading indicates the number of cases corresponding to each specific convective cloud fraction (CCF) and $PM_{2.5}$ bin. Note that $PM_{2.5}$ is separated into ten equal-sample bins. Black circles and error bars are the mean values and standard deviations of CCF in each $PM_{2.5}$ bin within each hour, magenta dots indicate the mean CCF over all times. Black solid lines represent the three-point moving average of the black circles. Red solid lines (with red numbers) mark the mean polluted and clean thresholds during each hour.




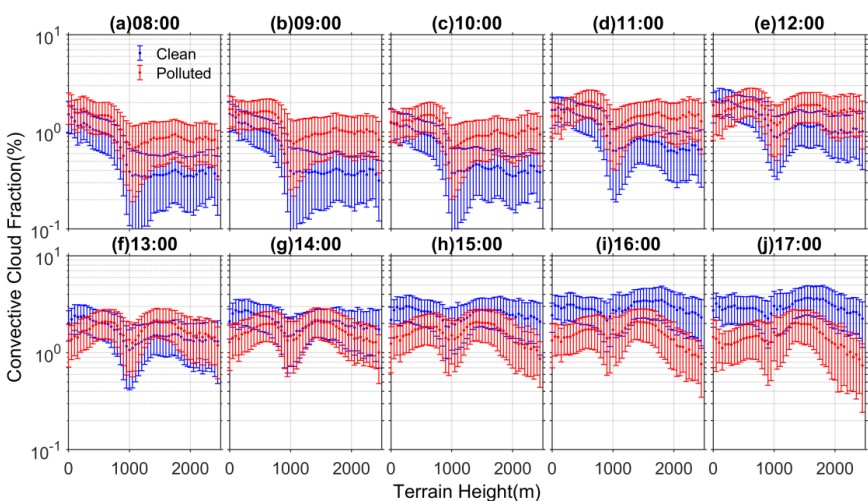


**Figure 9**. Convective cloud fraction (CCF) normalized by the total number of cases
with respect to terrain height changes during May-September, 2016-2017. Data for
polluted conditions are plotted in red, whereas those for clean conditions are shown in
blue. Dots and error bars are the mean values and standard deviations of the fractions
in each TH bin.



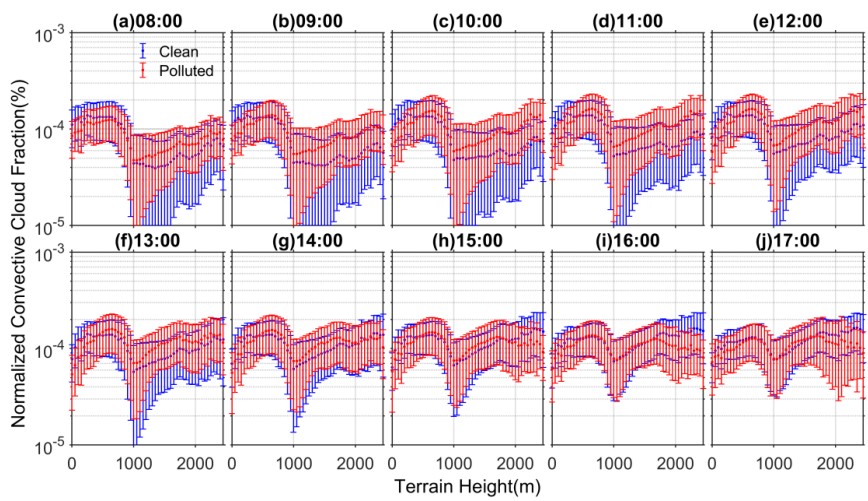

Figure 10. Same as Figure 9, but for CCF that is normalized by the number of cases under polluted and clean conditions during each hour, respectively (Equation 8).



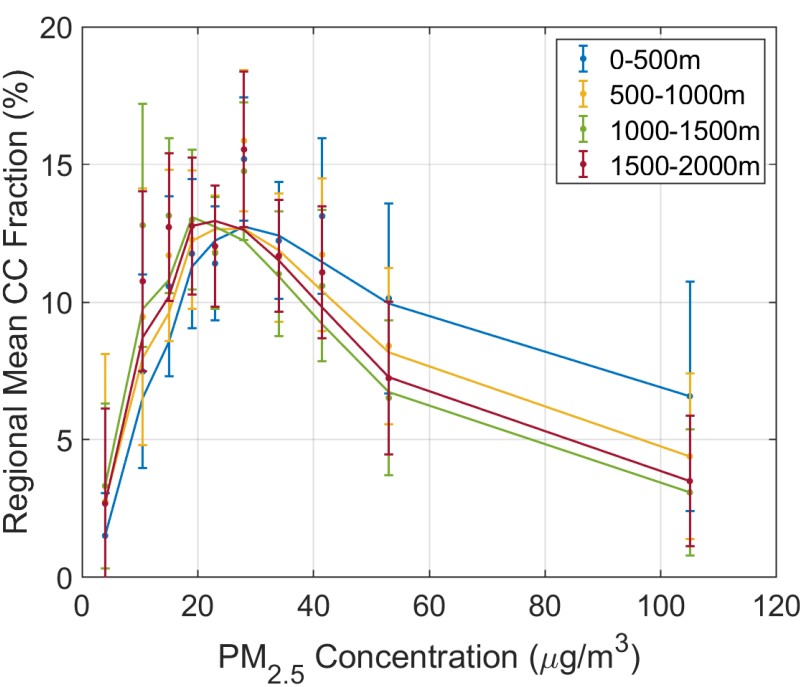

**Figure 11** Convective cloud fraction (CCF) over regions with terrain heights in the range 0-500 m (blue), 500-1000 m (yellow), 1000-1500 m (green) and 1500-2000 m (red) with respect to $PM_{2.5}$ concentration during May-September, 2016-2017. Ten equally sampled $PM_{2.5}$ bins are defined for each terrain height range. The standard deviations are shown as error bars. Each solid line represents the three-point moving average for dots in the corresponding color.



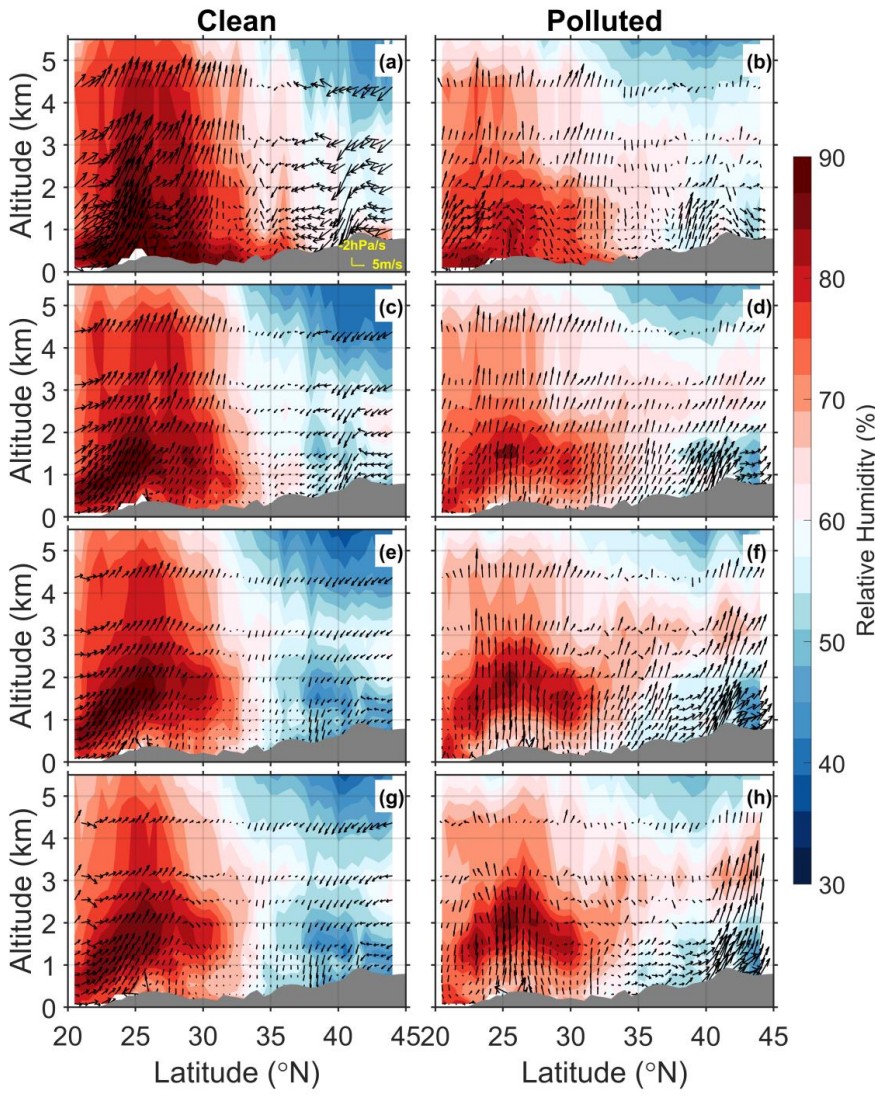

**Figure 12.** Latitude-altitude cross-sections of mean relative humidity (color-shaded)

and mean meridional-vertical wind (vectors) over the continent, averaged along 105 to

125°E for clean (left panel) and polluted (right panel) conditions at (a, b) 08:00 LT, (c,

d) 11:00 LT, (e, f) 14:00 LT and (g, h) 17:00 LT during May-September from 2016 to



2017. Vectors are constructed from the easterly wind (u) and vertical velocity (ω),

scaled with -100. Gray shaded parts are the meridional mean terrain heights within ROI.

1096

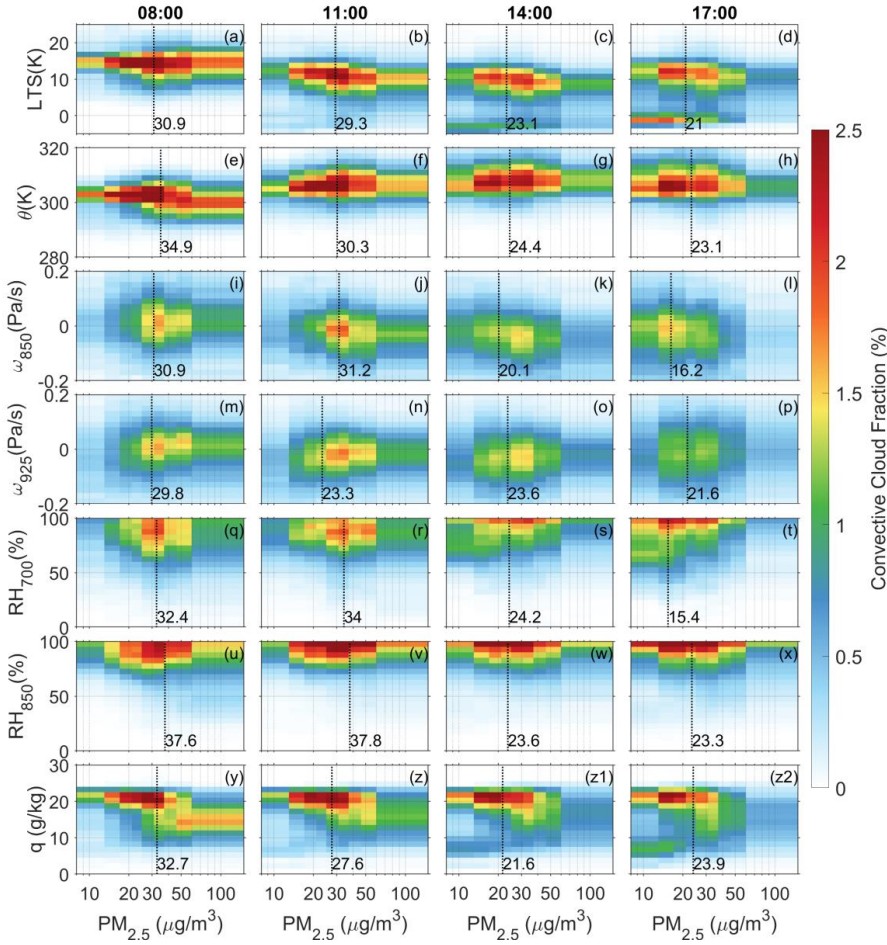

**Figure 13**. The joint distribution of convective cloud fraction (CCF) with respect to PM$_{2.5}$ concentration and (a-d) lower-tropospheric stability (LTS), (e-h) potential temperature ($\theta$), (i-l) vertical pressure velocity at 850 hPa ($\omega_{850}$) over terrain above 1000 m, (m-p) vertical pressure velocity at 925 hPa ($\omega_{925}$) over terrain below 1000 m, (q-t), relative humidity at 700 hPa (RH$_{700}$) over terrain above 1000 m, (u-x) relative humidity at 850 hPa (RH$_{850}$) over terrain below 1000 m, (y-z2) relative humidity at the surface (RH$_{surface}$). Each column represents a different local time during the day within the ROI, specifically at 08:00, 11:00, 14:00 and 17:00. Black dashed lines and the numbers



beside mark the mean tipping points of CCF at different thermodynamic, dynamical
and humidity levels. Note that x-axis is in log scale.





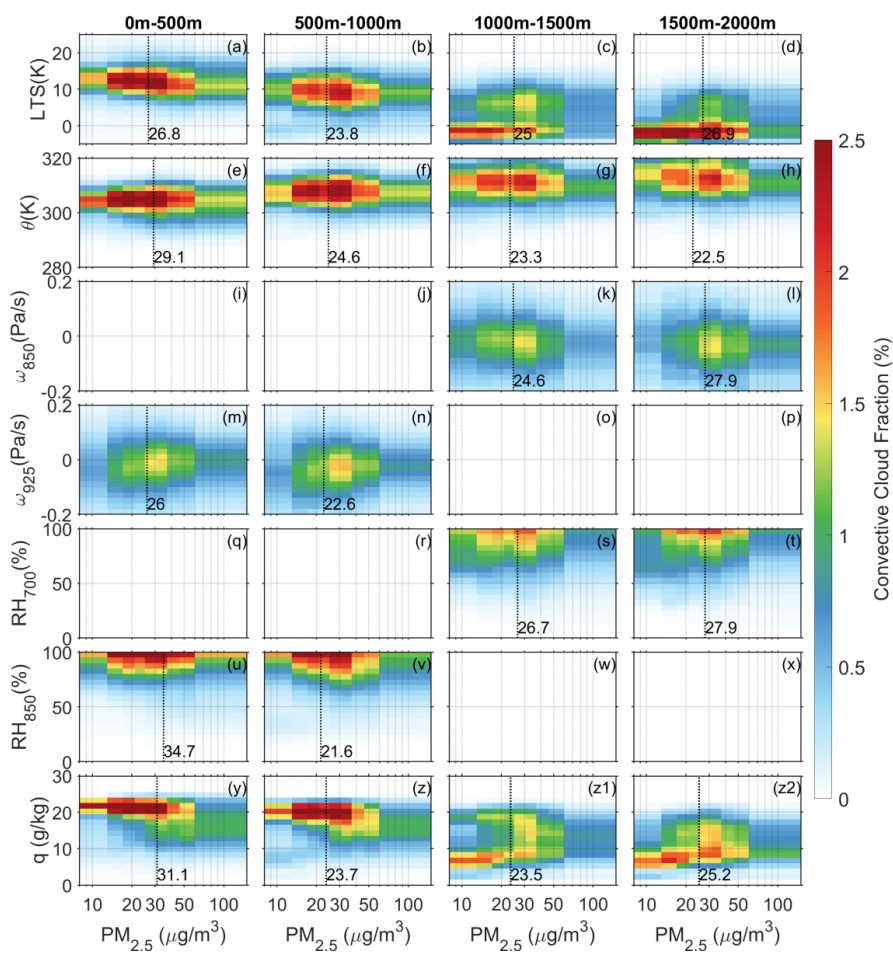


**Figure 14**. Same as Figure 13, but for different terrain heights; each column represents

a different terrain height range.


