# Peer review of "Potential impact of aerosols on convective clouds revealed by Himawari-8 observations over different terrain types in eastern China"

_Atmospheric Chemistry and Physics, 2020_

## Referee Comment (RC1) · Anonymous Referee #1 · 17 Nov 2020

**1   General comments**

Chen et al. present in their study "Potential impact of aerosols on convective clouds revealed by Himawari-8 observations over different terrain types in eastern China" the impact of terrain height, meteorological parameters, and diurnal cycle on the aerosol-cloud interaction. The authors used two seasons from May-September 2016-2017 of observations from the geostationary satellite Himawari to retrieve the cloud mask, particulate matter from measurement station to retrieve the pollution concentration, MERRA-2 reanalysis to retrieve lower-tropospheric stability, vertical velocity, potential temperature, relative humidity, and specific humidity at different levels in accordance

with the terrain height. The study describes an innovative convective cloud mask that the authors developed and they collocate spatially and temporally with pollution and meteorological information. They infer an exhaustive list of interaction of clouds with their environment with some important features on the aerosol-cloud interaction, convective clouds occur more likely under unstable environment with some caveat comparing the morning and the afternoon and they compared between clean and polluted environment for example. I specifically appreciated that the authors provide a physical explanation for the different observations they acknowledged that they are using observations and further work is needed to support the interpretations and conclusion.

The overall presentation is well structured and clear and the Figure are explanatory and provide argument for the text. The scientific method and assumptions seem valid and they are clearly outlined. I have some concerns regarding the conclusion of the article and the strong statement that the article study the impact of aerosol on convective clouds but I am not sure about that, as the data are not compared with clear sky situation, I developed what I mean in the next section. I have also some concern about the cloud mask validation, but I think it just a lake of details from the current version. Finally, working with a large amount of data, statistical tests are missing to quantify the effect observed. Otherwise, the topic and the results fit totally within the scope of ACP and I strongly recommend the publication in the ACP after some modifications.

**2   Main comments**

1. The study compares different regimes, handling a large dataset and the conclusion often belongs to observation from the Figures. Quantification through statistical tests are often missing to support the description and the conclusions. I put some examples here:

   - l. 394: The authors mention "non negligible contributor. . . ", the changes

in $\omega$ do not seem statistically significant, did the authors try to perform a statistical test?

- The key of the study is explained in lines 598-602, with the explanation of decorellation between meteorological parameters and the PM2.5 concentration. I do not know if if we can say that q (for example) is uncorrelated with PM2.5. Did the authors try to quantify the potential correlation between the different parameters?

- l. 537: The authors mention "especially before 14:00 LT", I do not see a difference at all between polluted and clean after 14:00 from Figure 10, I think the term "especially" should be change for "only". For this entire section, I doubt that the results are statistically significant, did the authors try to perform a test?

- Line 618: CCF peaks decrease under stronger updraft, I am not sure to understand what it is meant here, I think quantification would help.

- I am not convinced by Figure 11, the differences between the graphs are not high. Did the authors try to quantify the difference with a statistical test? Moreover, the highest terrain is not the one with the minimum CC fraction for PM2.5 greater than 30mug/m3. I am curious on how does it fit on the author's explanation? Is it the different between plateaus and hill?

2. I need some clarification about the cloud mask. l.295-l.307: Some of the thresholds from which the cloud mask is based on correspond sometime to mature and small convective clouds, is it a problem? Did the authors try to change the different thresholds? If yes, how does it affect the results?

The validation of the cloud mask has been performed on a specific day. The method works well on a day with a lot of convective clouds, but how does it perform if there are less convective clouds? When the authors compare with ISCCP regimes, does it refer to the cloud optical depth/cloud top pressure diagram? Can

the authors describe how the ISCCP define convective clouds? If I understand correctly, at 8:00 there is an accuracy of 20% of the author's cloud mask algorithm to detect convective clouds, is it correct? I consider the author's method more robust than the COT-CTP diagram. Can the authors comment on that? The results from this part are interesting but I am not convinced that it serves as a validation of their algorithm.

On Figure 3, there are some clouds that are not detected by the algorithm but they are by MODIS and they could be convective (on the eastern part of the figure), can the authors have an explanation for that and can they comment it? In the article, the authors mention the opposite and refer to other cloudy pixels. I think it mainly refer to cloud edges, can the authors comment on that?

3. I am confused about the aim and the conclusions of the study. On line 642 "However testing whether the results are due mainly to aerosol effects is only a first step", the authors answered many questions in the manuscript but I do not see how we can affirm that aerosols are the main contributor to convective cloud occurrence. Further analysis would be needed to study the aerosol-cloud interaction, comparing with non cloudy pixel occurrence constrained for meteorological parameters for example. I do not know if the meteorological conditions discussed in the text favored the convective cloud formation or "simply" the cloud formation. It is acknowledged in the text but it can be misleading in many parts of the manuscript.

How are the meteorological parameter variations with clear sky occurrence (for example Figure 5). Is it really the conditions which favored the convective cloud occurrence or is it the meteorological conditions difference between the areas and an other parameter which favor or inhibit the cloud formation? Do the authors take that into account?

4. Cloud interactions with aerosols and cloud processes are different over land or

over ocean. In the study the authors merged over open ocean and over land. Did the authors try to remove the ocean in their analysis?

**3  Specific comments**

1. Figure color bars: The rainbow color bar is not suited for colorblind people, I suggest to change it.

2. l. 37: I suggest to change to "convective cloud fraction increaseS then decreaseS"

3. l. 44: I suggest to change to "aerosolS decrease..."

4. l. 50: I suggest to change to "Convective cloudS are..."

5. l. 57: A space is missing between "climate" and (Zhao et...)

6. l. 60: I suggest to change to "light-absorbing aerosolS"

7. l. 103: "In recent years...", then the authors refer to Lynn et al. (2007), is it still considered as "recent years"

8. l. 105: "WRF" is not spelled out I think

9. l. 113: "Only few studies...", can the authors cite the few studies they are referring to.

10. section 2.1: It is not clear here if they consider data over ocean or not.

11. l. 160: Can the authors plot the region of interest in Figure 1, it is not clear which region is covered or not.

12. l. 188: Is PM$_1$ used in the study? The article mainly refer to PM$_{2.5}$ only.

13. l. 253: Is q really at the surface or at 2m above surface?

14. l. 271: What is the matrix size?

15. l. 300: "With mean contrast>3.5", where does the value of 3.5 comes from?

16. l.312: I suggest to change to "isolating convective cloudS"

17. Fig. S1: Is this graph only for July 30th 2016 or for the two seasons considered later?

18. Fig. 4: It is difficult to distinguish the gray line, the author should highlight it differently.

19. l. 398: "most common", do the authors mean "higher".

20. Fig. 4: Are the sub-figures snapshot of the specific time indicated or is it integrated over two hours?

21. l. 422-424: Considering the potentially high spatial variability of aerosol with the different terrain heights, how good is the "nearest-pixel" assumption?

22. Fig. 6: It is difficult to distinguish between the red and magenta lines, I suggest to find other colors.

23. l. 535: "We can see from Figure 11...", I think the authors want to refer to Figure 10.

24. l. 706: I suggest to change "convection by the enhancing" to "convection by enhancing"
25. l. 708: I suggest to change "development of convective cloud" to "development of convective cloudS"

26. Figure 7: There are red spots that the authors highlight with the presence of cities. What about the other red spots, is there any reason?

27. Figure 8: The caption mentions "red solid lines" but the lines seems dashed.

28. references: Many doi are missing

29. There are "Houze Jr, R. Q.: Cloud dynamics, Academic press" twice, for 1993 and 2014. Can the authors use only one edition ? I would suggest the most recent one.
* * *

---

## Referee Comment (RC2) · Anonymous Referee #2 · 24 Nov 2020

In this paper, the authors seek to explore correlations between aerosol and convective clouds over Eastern China, with a focus on how these correlations depend on meteorological factors and the diurnal cycle. They develop a new method for identifying convective clouds in geostationary satellite data that compares well with existing methods. Using surface measurements of PM2.5 as their CCN proxy, they show that the occurrence of convective cloud is correlated to the aerosol environment across a range of meteorological environments. This relationship varies depending on the time of day and the terrain altitude, which they suggest may indicate variations in the response of convective clouds to aerosol.

[Figure]

This paper makes a useful contribution, investigating and enumerating aerosol-cloud relationships in a variety of conditions. There are some things I would suggest modifying in the presentation and interpretation of the results, following which this paper may be suitable for publication in ACP.

**1   General points**

Correlation vs causation. This is mentioned at the end of the paper, but should probably be more clearly addressed throughout. In particular, it is implicitly assumed that the aerosol is independent of the meteorology (at least when presenting the aerosol-CCF results), but it is then clearly shown that PM2.5 is correlated with almost every meteorological variable studied (Fig. 13). The results in Fig. 13 are then used to interpret results as a changing sensitivity of CCF to aerosol, rather than the perhaps simpler conclusion that the observed relationships are driven by meteorology. The changes here do not have to be large, but the authors should make it clear when interpreting their results that they have shown a correlation between aerosol and cloud properties, rather than the aerosols are a cause of these relationships (and definitely not the main cause - L642).

I might also suggest changing "probable" to "potential" when discussing the TH relationship to aerosol-cloud correlations, as it has a slightly different meaning when considering the likelihood of such a relationship being due to aerosol.

Terrain height vs. gradient. The use of terrain height to try and identify some information about the in-cloud updraft is a very interesting route. I might suggest that the authors consider the gradient of the terrain, rather than the altitude. Global model parameterisations (e.g. Joos et al, JGR, 2008) typically rely on information about the terrain variability, rather than the absolute height, to infer updrafts. These parameterisations are intended for cirrus, but seem to provide some information on convective

cloud updrafts too (e.g. Gryspeerdt et al, ACP, 2018). Similarly, the proposed mechanism (Li and Weng, 1987, 1988, 1989) seems to point more towards terrain variability, rather than absolute magnitude - perhaps for study in future work.

References. I am sure that it is a simple oversight given the large number of references, but quite a high proportion of the references are written by one of the coauthors. The authors are experts in this area and have written many important papers, so this is to be expected. I wouldn't suggest removing anything at this stage (and these references are not inappropriate), but including papers on aerosol impacts on Arctic stratus while leaving out seminal papers on aerosol-cloud relationships in convective clouds (Koren et al, GRL, 2005 and Williams et al, JGR, 2002 being particularly notable) might suggest the references could use a bit of attention.

Connected to this, there is a large amount of work specifically around accounting for meteorological effects in aerosol-cloud relationships (following on from Koren et al., ACP, 2010; Quaas and Boucher, Nat. Geosci, 2012), which would be highly relevant for the interpretations of aerosol-cloud correlations in this work.

**2   Specific points**

L118 - they cannot track cloud development individually, there are a number of papers that use multiple polar-orbiting satellites to observe cloud development (e.g. Matsui et al., JGR, 2006; Meskhidze et al., ACP, 2009)

L189 - Is PM2.5 suitable for use as a CCN proxy? Presumably there has been some work done on this before, but I imagine the suitability as a proxy depends on aerosol type. Is that constant across this region, particularly between the lowlands and highlands?

L206 - A number of meteorological variables are quoted here, but different ones are

then described below. How do they relate to each other?

L215 - LTS has shown some skill at predicting stratiform cloud cover over the ocean (Klein and Hartmann, J. Clim, 1993). Is it really suitable for investigating convective clouds? It also has a dependence on the surface altitude, as the lapse rate is not usually the dry adiabatic lapse rate (e.g. Wood and Bretherton, J. Clim, 2006). Is it suitable for a study where the terrain height changes?

L230 - Wang et al (2018) are also using other meteorological variables that include humidity information (such as CAPE). CAPE is not used here (but might be suitable for looking at convective clouds?)

L243 - Lee et al (2019) is looking at surface heterogeneity in a high resolution model for a 28km domain. Is it a suitable reference relating the large scale average vertical velocity to cloud-scale updrafts?

L257 - This texture method looks like an interesting way to identify convective clouds, but I was not clear on what exactly is being done. Equations 3-6 are introduced, but apparently not then used or referred to. It is stated that the contrast at d=1 is used. Is this then the central-difference second derivative at each point? It would be good to have some way to visualize it for those not familiar with the GLCM.

L300 - Which wavelength is used to calculate the contrast?

L302 - It was stated earlier that the texture method could reduce the need for thresholds, but some are still included. How much extra does the texture filter add beyond these filters? L326 suggests that the majority of the difference from MOD35 comes from the cirrus filtering, as the TCT-CID method selects the convective cores.

L311 - "Numerically smaller " is a bit ambiguous. Do you mean "less than"?

L343 - Does this mean that the TCT-CID method has an error that depends on solar zenith angle? That might be an issue for looking at the diurnal cycle. Does it have a large impact?

L366 - Yang and Slingo, M. Weather Rev. (2001) might also be a useful reference, although it only covers the southern part of China

L371 - You probably only need to cite one edition of "Cloud Dynamics"

L373 - The gray lines were a bit difficult to see on my screen. Perhaps you could make them black and remove the provinces from the map?

L388 - The LTS has an in-built dependence on the above-inversion lapse rate and hence the altitude of the 700hPa level. This could explain part of the dependence on the altitude.

L396 - The split of RH and updraft levels for the high and low populations might generate an artificial difference between them. Might it be more appropriate to use an updraft and RH at a specific height above the surface instead?

L399 - Related to L396 - the difference may be that different variables are being compared? This make the following interpretation difficult.

L427 - Is there a regional bias in the clean/polluted pixels? Does this end up comparing N. China to S. China (for example)?

L451 - It is interesting that the change is first observed in smaller clouds and this is a useful result. I am not clear how aerosol-radiation interactions are responsible though.

L467 - An extra explanation step would be useful to help the reader. How does a decrease in surface temperature lead to a suppression of the vertical moisture flux?

L500 - This would be good to check.

L510 - I am not sure what the semicolon means here (rather than a comma)

L514 - This is a little bit ambiguous, is the daytime 0800-1700, or is it defined using solar zenith angle?

L526 - How are the values in Fig. 9 calculated? My reading of Eq. 8 would make it

constant. In Fig. 9, the values presented are averaged over the whole region, so

$$N\bar{C}CF \propto \Sigma_i\Sigma_j NCCF = \frac{\Sigma_i\Sigma_j N_{C(P)}(i,j,h,t)}{\Sigma_i\Sigma_j N_{C(P)}(i,j,h,t)} = \text{const.}$$

Clearly I am missing something here, as this would lead to the NCCF being constant in time and altitude.

L564 - How are clean conditions defined here? They were previously defined at a gridbox level, but this is apparently an average over the whole regions?

L568 - This is a good example of the correlation vs causation issue. While the reduced above cloud RH in polluted conditions could lead to a "restraint of convective cloud development due to the inhibition of (the) aerosol microphysical effect", a simpler explanation is that the reduced above cloud RH itself limits convection, which doesn't require an aerosol impact on cloud. It is not clear which explanation is correct (probably a little of both), but it is not clear to me that aerosols are responsible for the aerosol-CCF relationship.

L609 - "may prove", "are probably". I am not sure that the evidence is there to make this strong a claim. There is still a considerable correlation between the aerosol and meteorological environment (Fig. 13)

L619 - Updraft is also related to the activation - if this is an aerosol effect, an increased updraft can lead to a larger sensitivity to aerosol (moving to an aerosol-limited environment).

L642 - "testing whether the results are mainly due to aerosol effects" is the same things as establishing causality (as I read it). I would perhaps consider - "However, establishing a correlation between aerosol and cloud properties is only a first step;"

L690 - Perhaps the pattern is "...consistent with the combined action..."? Given the correlations between aerosol and meteorology (Fig. 13), it is not clear that these correlations imply causality. This follows in later parts of the conclusions. Correlations have

been shown (and these are very interesting!), but it is not yet clear these are the result of an aerosol impact on clouds.

**3 Additional references**

Boucher, O., and J. Quaas (2012), Water vapour affects both rain and aerosol optical depth, Nat. Geosci., 6(1), 4–5, doi:10.1038/ngeo1692.

Gryspeerdt, E., J. Quaas, T. Goren, D. Klocke, and M. Brueck (2018), An automated cirrus classification, Atmos. Chem. Phys., 18(9), 6157–6169, doi:10.5194/acp-18-6157-2018.

Joos, H., P. Spichtinger, U. Lohmann, J.-F. Gayet, and A. Minikin (2008), Orographic cirrus in the global climate model ECHAM5, J. Geophys. Res., 113, D18205, doi:10.1029/2007JD009605.

Koren, I., Y. J. Kaufman, D. Rosenfeld, L. A. Remer, and Y. Rudich (2005), Aerosol invigoration and restructuring of Atlantic convective clouds, Geophys. Res. Lett., 32, 14828, doi:10.1029/2005GL023187.

Koren, I., G. Feingold, and L. A. Remer (2010), The invigoration of deep convective clouds over the Atlantic: aerosol effect, meteorology or retrieval artifact?, Atmos. Chem. Phys., 10(18), 8855–8872, doi:10.5194/acp-10-8855-2010.

Meskhidze, N., L. A. Remer, S. Platnick, R. Negrón Juárez, A. M. Lichtenberger, and A. R. Aiyyer (2009), Exploring the differences in cloud properties observed by the Terra and Aqua MODIS Sensors, Atmos. Chem. Phys., 9(10), 3461–3475, doi:10.5194/acp-9-3461-2009.

Williams, E. et al. (2002), Contrasting convective regimes over the Amazon: Implications for cloud electrification, J. Geophys. Res., 107(D20), 8082,

doi:10.1029/2001JD000380.

Wood, R., and C. S. Bretherton (2006), On the Relationship between Stratiform Low Cloud Cover and Lower-Tropospheric Stability, J. Climate, 19(24), 6425–6432, doi:10.1175/JCLI3988.1.

Yang, G.-Y., and J. Slingo (2001), The Diurnal Cycle in the Tropics, M. Weather Rev., 129, 784, doi:10.1175/1520-0493(2001)129<0784:TDCITT>2.0.CO;2.
* * *

---

## Author Comment (AC1) · 4 Feb 2021

We thank the reviewers for their thoughtful and excellent comments and suggestions. We have tried as much as possible to address all concerns and have revised the manuscript accordingly. Please see the supplement file for the responses and the revised manuscript with track changes.

Please also note the supplement to this comment: https://acp.copernicus.org/preprints/acp-2020-845/acp-2020-845-AC1-supplement.zip

2020.

---

## Author Comment (AC2) · 4 Feb 2021

We thank the reviewers for their thoughtful and excellent comments and suggestions. We have tried as much as possible to address all concerns and have revised the manuscript accordingly. Please check the supplementary file for the responses and the revised manuscript with track changes.

Please also note the supplement to this comment: https://acp.copernicus.org/preprints/acp-2020-845/acp-2020-845-AC2-supplement.zip
* * *
[Figure]

2020.

---

## Author Response (AR1)

**Responses to Comments**

**Referee #1**

**1. General comments:**

Chen et al. present in their study "Potential impact of aerosols on convective clouds revealed by Himawari-8 observations over different terrain types in eastern China" the impact of terrain height, meteorological parameters, and diurnal cycle on the aerosol-cloud interaction. The authors used two seasons from May-September 2016-2017 of observations from the geostationary satellite Himawari to retrieve the cloud mask, particulate matter from measurement station to retrieve the pollution concentration, MERRA-2 reanalysis to retrieve lower-tropospheric stability, vertical velocity, potential temperature, relative humidity, and specific humidity at different levels in accordance with the terrain height. The study describes an innovative convective cloud mask that the authors developed and they collocate spatially and temporally with pollution and meteorological information. They infer an exhaustive list of interaction of clouds with their environment with some important features on the aerosol-cloud interaction, convective clouds occur more likely under unstable environment with some caveat comparing the morning and the afternoon and they compared between clean and polluted environment for example. I specifically appreciated that the authors provide a physical explanation for the different observations they acknowledged that they are using observations and further work is needed to support the interpretations and conclusion. The overall presentation is well structured and clear and the Figure are explanatory and provide argument for the text. The scientific method and assumptions seem valid and they are clearly outlined. I have some concerns regarding the conclusion of the article and the strong statement that the article study the impact of aerosol on convective clouds but I am not sure about that, as the data are not compared with clear sky situation, I developed what I mean in the next section. I have also some concern about the cloud mask validation, but I think it just a lake of details from the current version. Finally, working with a large amount of data, statistical tests are missing to quantify the effect observed. Otherwise, the topic and the results fit totally within the scope of ACP and I strongly recommend the publication in the ACP after some modifications.

**We thank the reviewers for their thoughtful and excellent comments and suggestions. We have tried as much as possible to address all concerns and have revised the manuscript accordingly. The reviewers' comments are written in normal font, and our point-to-point responses to the reviewers' comments are in bold.**

**2. Main comments**

1. The study compares different regimes, handling a large dataset and the conclusion often belongs to observation from the Figures. Quantification through statistical tests are often missing to support the description and the conclusions. I put some examples here:

• l. 394: The authors mention "non negligible contributor. . . ", the changes in ω do not seem statistically significant, did the authors try to perform a statistical test?

**Response: By considering main comment 3), we have changed Figure 5 into the CC OF anomaly, and carried out a $\chi^2$-test to see if the differences are significant. More details can be found in the response to main comment 3).**

• The key of the study is explained in lines 598-602, with the explanation of de-correlation between meteorological parameters and the PM2.5 concentration. I do not know if we can say that q (for example) is uncorrelated with PM2.5. Did the authors try to quantify the potential correlation between the different parameters?

**Response: By examining all potential influences, we found that $PM_{2.5}$ concentrations can have correlations with meteorological factors. So, we tried to plot the 2-d CCF distribution under different meteorological parameters and different $PM_{2.5}$ concentrations. We have divided the $PM_{2.5}$ concentration into 10 equally sampled bins to reduce the influence of sampling bias for the CCF distribution. Thus, although we cannot exclude the influence of meteorological factors to $PM_{2.5}$, the relationship between aerosol and convective cloud is still robust.**

**Additionally, to avoid the impact of hydration on aerosol, the PM2.5 measurements we used are dried particle masses; we have added this statement in section 2.2 and 4.4.**

**In section 2.2:** *"On the other hand, particulate matter can be measured from the surface or aircraft under all-sky conditions and can provide dried particle masses which can minimize the influence of moisture on it."*

**In section 4.4:** *"As PM2.5 measurements are dried particle masses, which can minimize the influence of ambient moisture, these patterns are more likely related to the strengthening of the aerosol microphysical effect, which might overtake the suppression from the aerosol radiative effect in higher aerosol loading conditions in these regions."*

**We also added a statement in section 5 to further explain the relationship of meteorological factors and aerosol effects on convective clouds:**

*"A possible alternative explanation is that aerosols and CCF are associated because they are both affected simultaneously by the same meteorological factors. However, CCF had generally similar relationships with aerosol mass concentrations for all meteorological stratifications that were examined. This observation renders the alternative meteorological explanation less likely."*

• l. 537: The authors mention "especially before 14:00 LT", I do not see a difference at all between polluted and clean after 14:00 from Figure 10, I think the term "especially" should be change for "only". For this entire section, I doubt that the results are statistically significant,

did the authors try to perform a test?

**Response: We have removed the word "especially" in this sentence, and we have also replotted Figure 9 and 10 to make the mean value more visible and added dots on the points that have differences exceeding the 95% significance level according to a student's t test.**

• Line 618: CCF peaks decrease under stronger updraft, I am not sure to understand what it is meant here, I think quantification would help.

**Response: Thanks for pointing out the problem, here we only mean that the turning point is moving to smaller values, not the CCFs, we have fixed this sentence as follows:**

*"However, the turning values of CCF generally decrease with increasing updraft conditions as well"*

• I am not convinced by Figure 11, the differences between the graphs are not high. Did the authors try to quantify the difference with a statistical test? Moreover, the highest terrain is not the one with the minimum CC fraction for PM2.5 greater than $30\mu g/m^3$. I am curious on how does it fit on the author's explanation? Is it the different between plateaus and hill?

**Response: Per your suggestion, we did significant tests for all the points in the figure. Within each PM2.5 bin, Student's t-tests were carried out for each pair of CC fractions for the four terrain height bins. For all the distributions, the null hypotheses are rejected, which means that each pair of distributions is significantly different (at 95% significance level). The p-values of the t-test for the ten $PM_{2.5}$ bins are listed as follows:**

**Tabel R1. p-values of the t-test for the ten $PM_{2.5}$ bins to each pair of the terrain-height bins. Note that differences that passes the 95% significance level are in bold.**

| Terrain Heights (m) | | PM2.5 bins | | | | | | | | | |
|---|---|---|---|---|---|---|---|---|---|---|---|
| Group1 | Group2 | 1 | 2 | 3 | 4 | 5 | 6 | 7 | 8 | 9 | 10 |
| 0-500 | 500-1000 | **0.00** | **0.00** | **0.00** | **0.00** | **0.00** | **0.00** | **0.00** | **0.00** | **0.00** | **0.00** |
| 0-500 | 1000-1500 | **0.00** | **0.00** | **0.00** | **0.00** | **0.00** | **0.00** | **0.00** | **0.00** | **0.00** | **0.00** |
| 0-500 | 1500-2000 | **0.00** | **0.00** | **0.00** | **0.00** | **0.00** | **0.00** | **0.00** | **0.00** | **0.00** | **0.00** |
| 500-1000 | 1000-1500 | **0.00** | **0.00** | **0.00** | **0.00** | **0.01** | **0.00** | **0.00** | **0.00** | **0.00** | **0.00** |
| 500-1000 | 1500-2000 | 0.10 | **0.00** | **0.00** | **0.00** | **0.00** | **0.00** | **0.01** | **0.00** | **0.00** | **0.00** |
| 1000-1500 | 1500-2000 | **0.00** | **0.00** | **0.00** | **0.00** | **0.00** | **0.00** | **0.00** | **0.00** | **0.00** | **0.00** |

**Meanwhile, each of the CCF curves is calculated within each sub-region, which means that the sum of all the numbers of each CCF curve is 100%. So, we rewrote the sentence as follows:**

*"We calculate the CCF using the number of convective clouds within each of 10 equally sampled PM2.5 bins divided by the total convective cloud amount over each sub-region. Each pair of CCFs in the four sub-regions is significantly different at 95% significance level according to the Student's t-test."*

**Over higher terrains, aerosol concentrations are relatively lower, and the effect of topography might play a more important role. Surface heating is stronger at the higher elevation, which may dominate the trend of decreasing CCF by more convective cloud**

**formation, so that the inhibition of convective cloud by aerosol radiative effect is weaker. Thus, the CCF at PM$_{2.5}$ greater than 30 μg/m$^3$ over 1500-2000m region could be larger than CCF over 1000-1500m region.**

    **We have also added this explanation in the context:**

    ***"Over higher terrains (TH>1000m), aerosol concentrations are relatively low, and the effect of topography might play a more important role. Surface heating is relatively stronger at the high mountains(TH>1500m), which may dominate the trend of decreasing CCF after the turning zone, so that the inhibition of convective cloud by aerosol radiative effect is weaker, and the CCF is higher than the plateau regions (1000<TH≤1500m) after the turning zone."***

2. I need some clarification about the cloud mask.

- l.295-l.307: Some of the thresholds from which the cloud mask is based on correspond sometime to mature and small convective clouds, is it a problem? Did the authors try to change the different thresholds? If yes, how does it affect the results?

**Response: One of the objectives of this study is to find out the relationship between aerosols and convective clouds, as we have set a threshold to select convective cloud smaller than 10000 pixels, mostly small local-scale convective cloud systems are included. We combined the small and mature convective cloud systems only to see whether aerosol can affect the occurrence of convective clouds, but did not draw much attention to the aerosol effect on different stages of development or types of convective clouds. We assume that aerosol loading can affect the triggering and development of convective clouds, so that the occurrence frequencies are relatively different under polluted and clean conditions regardless of their stage of development. We have also classified the identified convective clouds based on their cloud top temperatures being above or below 0°C as shallow or deep convection, and found that their responses to aerosol are similar (Figure R1 and R2), with more convective clouds found before noon time and less in the afternoon under polluted conditions in general. These results further indicate that the occurrence of small local-scale convective clouds can be influenced by aerosol loading regardless of the development stage, although the effect may alter the development of convective clouds. So, the results we used in this study have just combined together the shallow and deep convective clouds.**

[Figure]

**Figure R1. Diurnal changes of shallow convective cloud (mean cloud top temperature > 0°C) fraction difference between polluted and clean environments (Polluted-Clean) during May-September in 2016-2017. Time marked above each figure is the local time. Grid points are plotted only if they exceed the 95% significance level (p < 0.05) according to the Pearson's χ2 test.**

[Figure]

**Figure R2. Same as Figure R1, but for deep convective clouds (mean cloud top temperature < 0°C)**

However, there are differences in detail between the shallow and deep convection. The reason for these differences is likely to lie in the different effects of aerosol at different stages of convective cloud development. So, we are planning to explore the responses of convective clouds at different stages of development to aerosol loading in future work, so as to provide more details about the changes in convective clouds under polluted conditions.

As the thresholds we use in this study are mainly from VIS channel, it is certain that the areas of identified cloud clusters can vary with the choice of threshold. But we think the main difference may lie in the ratio of deep to shallow cloud clusters, and as these filters become stricter, most of the mature convective cloud anvil and some very shallow convection will be excluded. But, as we previously found, the response of shallow and deep convective cloud to aerosol loading could be similar. Therefore, the changes in the thresholds may not have very large impact on the results.

- The validation of the cloud mask has been performed on a specific day. The method works well on a day with a lot of convective clouds, but how does it perform if there are less convective clouds? When the authors compare with ISCCP regimes, does it refer to the cloud optical depth/cloud top pressure diagram? Can the authors describe how the ISCCP define convective clouds? If I understand correctly, at 8:00 there is an accuracy of 20% of the author's cloud mask algorithm to detect convective clouds, is it correct? I consider the author's method more robust than the COT-CTP diagram. Can the authors comment on that? The results from this part are interesting but I am not convinced that it serves as a validation of their algorithm.

**Response: Figure 3 is only taken as an example of identified convective clouds to show the performance of the TCT-CID method, but our validation employs much more data (about 20455 scenes from May to September in 2016 and 2017) in Figure S2. To show the performance of the cloud identification method more clearly, we have replotted Figure 3 in smaller area, and added another case comparison between MODIS cloud mask and our identification result. We choose a scene with sparse convective cloud in the mountain area in northern China and nearly no other types of clouds are included. We have rephrased the sentences to make this point clearer as follows:**

*"Figure 3 presents two examples of convective clouds identified in the hilly regions in southern China at 13:40 LT on July 30th, 2016 and in the mountain regions in northern China at 14:20 LT on June 22nd, 2016. To get a general idea of the performance of the TCT-CID method, we compare the identified convective cloud masks against the MODIS/Aqua MYD35 cloud masks. As the MODIS product does not classify clouds into different types, a scene which contains a vast convective cloud field and a scene with sparse convective clouds are chosen."*

**In Figure S2, we plot the ratio of ISCCP cloud types corresponding to the identified convective cloud, which means that the frequencies of different cloud types corresponding to the identified convective cloud masks are calculated as the percentage of the pixel number of convective cloud mask which matches each ISCCP cloud type to all the cloud mask pixels. For example, at 08:00, over 20% of the identified convective cloud mask matches the level 2 DCC cloud type. To make this point clearer, we have rewritten this paragraph in section 3.2 as follows:**

*"This product provides the cloud type information using the ISCCP cloud classification criteria, which defines cloud type using specific combinations of cloud top pressure (CTP) and cloud optical thickness (COT) (Rossow and Schiffer, 1991). A COT-CTP diagram is created to differentiate cloud types with different radiative feedbacks. This method provides*

*to some degree an accurate identification and classification of various cloud types. In this study, we compared the identified convective cloud masks with the L2CLP cloud type product. The frequencies of different cloud types corresponding to the identified convective cloud masks are calculated as the percentage of the convective cloud mask pixel count that matches each ISCCP cloud type to all the cloud mask pixels (Figure S2)."*

*Reference:*

*Rossow, W. B., and Schiffer, R. A.: ISCCP Cloud Data Products, 10.1175/1520-0477(1991)0722.0.CO;2, 1991.*

We provide an example of a case comparison between our identification method and the L2CLP cloud type product. The red part in Figure R3b is the identified convective cloud by our method, the cyan part is the L2 cloud type product (Only deep convective cloud, cumulus and stratocumulus are included), the blue parts are the overlaps of the two products. We can find a generally good agreement with the two products. Our method shows better performance in identifying the deep convective clouds, the areas of the deep convections are larger than the L2 product, but the capability of identifying shallow convective cloud is limited. We also added a discussion in section 3.2 to point this out:

*"The method is designed to find sharp edges and bright clusters in VIS images firstly, but as some of the high clouds may also fit these conditions, which are not the subject of this study, we added a split-window filter to exclude this kind of clouds. However, the addition of this filter will also exclude the anvils of some mature convective clouds and some of the shallow clouds, so the cloud area identified by our method is smaller than the cloudy area found by the MODIS cloud mask and the capability of shallow convection identification is limited. Nevertheless, the majority of convective clouds are well captured by our method."*

[Figure]

**Figure R2. Comparison between convective cloud mask identified by our method and the Himawari-8 level 2 cloud type product. (a) True color image, (b) comparison between the two products, red parts are convective cloud mask from our method, cyan are from L2 product including DCC, Cu and Sc, blue parts are the overlaps.**

- On Figure 3, there are some clouds that are not detected by the algorithm but they are by

MODIS and they could be convective (on the eastern part of the figure), can the authors have an explanation for that and can they comment it? In the article, the authors mention the opposite and refer to other cloudy pixels. I think it mainly refer to cloud edges, can the authors comment on that?

**Response: By rechecking the data, we amended the figure of MODIS cloud mask in Figure 3. As we have screened the cirrus with the split-window method to isolate only convective clouds, the cloud area identified by our method is smaller than the cloudy area found by the MODIS cloud mask.**

3. I am confused about the aim and the conclusions of the study. On line 642 "However testing whether the results are due mainly to aerosol effects is only a first step", the authors answered many questions in the manuscript but I do not see how we can affirm that aerosols are the main contributor to convective cloud occurrence. Further analysis would be needed to study the aerosol-cloud interaction, comparing with non-cloudy pixel occurrence constrained for meteorological parameters for example. I do not know if the meteorological conditions discussed in the text favored the convective cloud formation or "simply" the cloud formation. It is acknowledged in the text but it can be misleading in many parts of the manuscript. How are the meteorological parameter variations with clear sky occurrence (for example Figure 5)? Is it really the conditions which favored the convective cloud occurrence or is it the meteorological conditions difference between the areas and another parameter which favor or inhibit the cloud formation? Do the authors take that into account?

**Response: We appreciate this comment. We would admit that there are limitations in reaching definite conclusions only based on observational data in the current study. Thus, we have rephrased many of the sentences in sections 4.2-4.4 and the summary to make it clear that we cannot find aerosol "as the reason" of the phenomena but is likely "to be related" to the phenomena. And we have changed the sentence you mentioned here to:**

*"However, identifying the apparent correlations between convective cloud and aerosol loading is only a first step, supporting the hypothesized mechanisms by which aerosol affects convective cloud occurrence. The mechanism by which the aerosol and meteorology interact is another important question, that is beyond the scope of the current study."*

**To compare the convective cloudy pixels with the non-cloudy pixels, we have replotted Figure 5. Instead of the using the convective cloud occurrence frequency (CC OF), we use the anomaly of CC OF. This could have made the difference in meteorological conditions between the presence of convective cloud and the all-sky conditions clearer. And we also rewrote section 4.1 to describe the new Figure 5. Generally, we can find from Figure 5 that convective clouds prefer stronger CAPE, surface heating, updraft and higher moisture, and the difference in mean values between the convective-cloud condition and the all-sky condition is larger over higher terrain regions.**

**There are differences in the meteorological conditions between different areas, especially for lower and higher terrains, and differences are also obvious when convective cloud occur. Thus, we generate Figure 13 and 14 to discuss the potential influences of meteorological parameters to the aerosol-CCF relationships.**

4. Cloud interactions with aerosols and cloud processes are different over land or over ocean. In the study the authors merged over open ocean and over land. Did the authors try to remove the ocean in their analysis?

**Response: In this study, the relationship between aerosol and convective cloud is mainly discussed over land, as the PM2.5 observations are available only over the continent. The ocean area is removed from the study. To make this point clear, we have added a statement in section 2.1 as follows:**

**"As PM2.5 data is only available over mainland China in this area, the aerosol-convective-cloud relationship is only discussed over the continent in this study."**

**Additionally, we have also removed the data over ocean area in Fig 7.**

**3. Specific comments**

1. Figure color bars: The rainbow color bar is not suited for colorblind people; I suggest to change it.

**Response: Thanks for the suggestion, it is great to consider the readers, we have changed the rainbow color in Figure 4, 8, 13 and 14 with a simpler color scale.**

2. l. 37: I suggest to change to "convective cloud fraction increaseS then decreaseS"

**Response: Changed.**

3. l. 44: I suggest to change to "aerosolS decrease. . . "

**Response: Changed.**

4. l. 50: I suggest to change to "Convective cloudS are. . . "

**Response: Changed.**

5. l. 57: A space is missing between "climate" and (Zhao et. . . )

**Response: Added.**

6. l. 60: I suggest to change to "light-absorbing aerosolS"

**Response: Changed.**

7. l. 103: "In recent years. . . ", then the authors refer to Lynn et al. (2007), is it still considered as "recent years"

**Response: We have removed the statement "In recent years" in this paragraph.**

8. l. 105: "WRF" is not spelled out I think

**Response: We have added the full name of WRF as "Weather Research and Forecasting (WRF) Model".**

9. l. 113: "Only few studies. . . ", can the authors cite the few studies they are referring to.

**Response: We have rephrased the sentence and added references here:**

*"Only a few studies include long-term observational data to analyze the relationships between aerosol and orographic precipitation statistically (e.g. Rosenfeld, 2007; Guo et al., 2014)"*

*References:*
*Rosenfeld, D., Jin Dai, Xing Yu, Zhanyu Yao, Xiaohong Xu, Xing Yang, Chuanli Du (2007), Inverse Relations Between Amounts of Air Pollution and Orographic Precipitation, Science, 315, 1396-1398, doi:10.1126/science.1137949.*
*Guo, J., M. Deng, J. Fan, Z. Li, Q. Chen, P. Zhai, Z. Dai, and X. Li (2014), Precipitation and air pollution at mountain and plain stations in northern China: Insights gained from observations and modeling, 119(8), 4793-4807, doi:https://doi.org/10.1002/2013JD021161.*

10. section 2.1: It is not clear here if they consider data over ocean or not.
**Response: Only convective cloud over land is considered in this study, to make it clearer, we have added a statement in this section.**

*"As PM$_{2.5}$ observations are available only over land in this area, the aerosol-convective-cloud relationship is only discussed over land in this study."*

11. l. 160: Can the authors plot the region of interest in Figure 1, it is not clear which region is covered or not.
**Response: The whole region in Figure 1 is the region of interest in this study, we have rephrased these sentences in section 2.1 as follows:**

*"In order to investigate the joint impact of aerosol pollution and topography on convective cloud fraction and diurnal variation, we chose the area within longitudes 105°E to 125°E, and latitudes 20°N to 45°N as the region of interest (ROI) for this study. We show the terrain distribution and the mean concentration of particles with aerodynamic diameters smaller than 2.5 μm (PM$_{2.5}$) during May-September in 2016-2017 over the ROI in eastern China in Figure 1. Generally, terrain height (TH) tends to increase from east to west in this region, and PM$_{2.5}$ mass concentration is generally higher over the plains and lower over mountain ranges and plateaus."*

12. l. 188: Is PM1 used in the study? The article mainly refer to PM2.5 only.
**Response: PM$_1$ is not used in this study; here we mean that particulate matter with small diameter, such as PM$_1$ or PM$_{2.5}$, may be taken as a proxy of CCN. To make the statement clearer, we have rephrased this sentence as follows:**

*"Particle size up to 10 μm may be much larger than the typical scale of CCN, so particulate matter up to 1 μm (PM$_1$) or 2.5 μm (PM$_{2.5}$) in diameter is more appropriate to serve as a CCN proxy. Due to the limited availability of PM$_1$ measurements in eastern China, we chose PM2.5 as an indicator of different CCN levels in the environment for this study."*

13. l. 253: Is q really at the surface or at 2m above surface?
**Response: Specific humidity at 2 m above the surface is used to represent the near-surface moisture. We have changed the "surface specific humidity" to "specific humidity at 2m".**

14. l. 271: What is the matrix size?

**Response: In this study, the matrix size is the size of the image in the region of interest; there are 1251×1001 pixels in the region. To make it clear, we have added *"(in this study, m = 1251, and n = 1001, as both the zonal and meridional spatial resolutions are 0.02° in the ROI)"* in the text.**

15. l. 300: "With mean contrast>3.5", where does the value of 3.5 comes from?

**Response: The clustering method provides us five clusters, as the identification tends to be unstable when the cluster number larger than 5. And we exclude the two clusters with lowest mean contrast, the second smallest mean value is about 3.5. We rephrased the sentence as follows:**

*"Five clusters are classified, and the mean contrast values are calculated. Those clusters with relatively higher "contrast" (with mean contrast>3.5, which is the second smallest among all the cluster mean contrast values) are considered either small convective clouds or the edges of mature convective clouds."*

16. l.312: I suggest to change to "isolating convective cloudS"

**Response: Changed.**

17. Fig. S1: Is this graph only for July 30th 2016 or for the two seasons considered later?

**Response: This figure is for the two seasons (from May to September in both 2016 and 2017). We have added the time period in the figure caption as follows:**

*"Figure S1. Frequencies of cloud types corresponding to the identified convective cloud with TCT-CID method in the warm seasons (May to September) of 2016 and 2017…."*

18. Fig. 4: It is difficult to distinguish the gray line, the author should highlight it differently.

**Response: As the gray line is hard to see in this figure, we have removed the province boundaries and plot the gray line in black.**

19. l. 398: "most common", do the authors mean "higher".

**Response: We have changed the word into "higher".**

20. Fig. 4: Are the sub-figures snapshot of the specific time indicated or is it integrated over two hours?

**Response: The subfigures are calculated by accumulating all the 6 observations (every 10min) within the hour (e.g., from 08:00-08:50) during the two warm seasons of 2016 and 2017. To make this point clearer, we have added a statement in section 4.1.**

*"Figure 4 shows the frequency of convective clouds occurring between 08:00 and 17:00 LT, calculated by dividing the number of convective clouds observed in six observations per hour (one every 10 minutes, e.g., 08:00-08:50) by the accumulation between 08:00 and 17:00."*

21. l. 422-424: Considering the potentially high spatial variability of aerosol with the different terrain heights, how good is the "nearest-pixel" assumption?

**Response: By considering this comment, we have made a comparison between the station-measured and the interpolated PM$_{2.5}$ concentrations in the following figure. The scattered dots are the PM$_{2.5}$ measurements at each site, and the gray dots are the interpolated PM$_{2.5}$ at each grid box. We can find very good agreement between these two datasets, which indicates that the assumption did not bring large uncertainties to the spatial distribution of PM$_{2.5}$.**

[Figure]

**Figure R3. Station-measured PM$_{2.5}$ (red dots) and the interpolated PM$_{2.5}$ (gray dots). The linear regression lines are shown as solid red line for the station-measured PM$_{2.5}$ and the dashed black line for the interpolated PM$_{2.5}$, respectively.**

22. Fig. 6: It is difficult to distinguish between the red and magenta lines, I suggest to find other colors.
**Response: Per your suggestion, we have changed it into a solid red line to make it clearer for readers to see.**

23. l. 535: "We can see from Figure 11. . . ", I think the authors want to refer to Figure 10.
**Response: Thanks for pointing out the mistake, the mistake has been corrected.**

24. l. 706: I suggest to change "convection by the enhancing" to "convection by enhancing".
**Response: Changed.**

25. l. 708: I suggest to change "development of convective cloud" to "development of convective cloudS"
**Response: Changed.**

26. Figure 7: There are red spots that the authors highlight with the presence of cities. What about the other red spots, is there any reason?
**Response: We have discussed other red spots in the text, to make it clearer, we have**

**rephrased the sentence as follows:**

*"There are also red dots located near several mountain areas, as complex topography may also be related to such a phenomenon. Furthermore, different topography may also lead to different convective cloud response to aerosol loading."*

27. Figure 8: The caption mentions "red solid lines" but the lines seem dashed.
**Response: We have replotted Figure 8 to change the color as concerned in comment 1), and increase the linewidth of the red lines.**

28. references: Many doi are missing
**Response: We have added dois for the references as much as we can.**

29. There are "Houze Jr, R. Q.: Cloud dynamics, Academic press" twice, for 1993 and 2014. Can the authors use only one edition ? I would suggest the most recent one.
**Response: We have removed the 1993 edition.**

**Referee #2**

In this paper, the authors seek to explore correlations between aerosol and convective clouds over Eastern China, with a focus on how these correlations depend on meteorological factors and the diurnal cycle. They develop a new method for identifying convective clouds in geostationary satellite data that compares well with existing methods. Using surface measurements of PM2.5 as their CCN proxy, they show that the occurrence of convective cloud is correlated to the aerosol environment across a range of meteorological environments. This relationship varies depending on the time of day and the terrain altitude, which they suggest may indicate variations in the response of convective clouds to aerosol.

This paper makes a useful contribution, investigating and enumerating aerosol-cloud relationships in a variety of conditions. There are some things I would suggest modifying in the presentation and interpretation of the results, following which this paper may be suitable for publication in ACP.

**We thank the reviewers for their thoughtful and excellent comments and suggestions. We have tried as much as possible to address all concerns and have revised the manuscript accordingly. The editor and reviewers' comments are written in normal font, and our point-to-point responses to the reviewers' comments are in bold.**

**1. General points**

1) Correlation vs causation. This is mentioned at the end of the paper, but should probably be more clearly addressed throughout. In particular, it is implicitly assumed that the aerosol is independent of the meteorology (at least when presenting the aerosol-CCF results), but it is then clearly shown that PM2.5 is correlated with almost every meteorological variable studied (Fig. 13). The results in Fig. 13 are then used to interpret results as a changing sensitivity of CCF to aerosol, rather than the perhaps simpler conclusion that the observed relationships are driven by meteorology. The changes here do not have to be large, but the authors should make it clear when interpreting their results that they have shown a correlation between aerosol and cloud properties, rather than the aerosols are a cause of these relationships (and definitely not the main cause - L642).

   **Response: We appreciate this comment. We have rewritten many parts of the figure descriptions for Figure 7-14. Most of the details can be found in section 4.2-4.4 and the Summary. Also, some points can be seen in responses to specific comments 24)-28).**

   **We also added a statement in section 5 to further explain the relationship of meteorological factors and aerosol effects on convective clouds:**

   ***"A possible alternative explanation is that aerosols and CCF are associated because they are both affected simultaneously by the same meteorological factors. However, CCF had generally similar relationships with aerosol mass concentrations for all meteorological stratifications that were examined. This observation renders the alternative***

*meteorological explanation less likely."*

2) I might also suggest changing "probable" to "potential" when discussing the TH relationship to aerosol-cloud correlations, as it has a slightly different meaning when considering the likelihood of such a relationship being due to aerosol.

   **Response: Thank you for the suggestion, we have changed the word "probable" to "potential". And we also have rephrased several parts in discussing the TH influences. Details can be found in section 4.3 ,4.4 and the summary.**

3) Terrain height vs. gradient. The use of terrain height to try and identify some information about the in-cloud updraft is a very interesting route. I might suggest that the authors consider the gradient of the terrain, rather than the altitude. Global model parameterizations (e.g. Joos et al, JGR, 2008) typically rely on information about the terrain variability, rather than the absolute height, to infer updrafts. These parameterizations are intended for cirrus, but seem to provide some information on convective cloud updrafts too (e.g. Gryspeerdt et al, ACP, 2018). Similarly, the proposed mechanism (Li and Weng, 1987, 1988, 1989) seems to point more towards terrain variability, rather than absolute magnitude - perhaps for study in future work.

   **Response: We would admit that terrain height and terrain variability can affect meteorology differently. However, for the study region, the height and variability are generally correlated. We calculate slope as the angle between the slope surface and the horizontal plane for each pixel, and found that the frequencies generally decrease with terrain heights within the range of 0-5° but increase within the 5°-10° bin, indicating that higher terrains generally have larger slopes (Figure R1) at least in the study region.**

   **In this study, we also use the terrain heights to represent plains (0-500m), hills (500-1000m), plateaus(1000-1500m) and high mountains(1500-2000m). From Figure R1, we found that both the plains and plateaus are topographically smooth with a slope less than 5° for more than 80% of the regions, and the slopes of hills and high mountain are relatively larger, which conforms to the basic features of these types of terrains. Therefore, using terrain height to represent different topography is basically a valid way.**

[Figure]

**Figure R1. Slopes of the terrain in the ROI. Regions are divided into 4 bins by terrain heights.**

4) References. I am sure that it is a simple oversight given the large number of references, but quite a high proportion of the references are written by one of the coauthors. The authors are experts in this area and have written many important papers, so this is to be expected. I wouldn't suggest removing anything at this stage (and these references are not inappropriate), but including papers on aerosol impacts on Arctic stratus while leaving out seminal papers on aerosol-cloud relationships in convective clouds (Koren et al, GRL, 2005 and Williams et al, JGR, 2002 being particularly notable) might suggest the references could use a bit of attention.

Connected to this, there is a large amount of work specifically around accounting for meteorological effects in aerosol-cloud relationships (following on from Koren et al., ACP, 2010; Quaas and Boucher, Nat. Geosci, 2012), which would be highly relevant for the interpretations of aerosol-cloud correlations in this work.

**Response: Per your suggestion, we have added these references in the revised manuscript accordingly.**

**2. Specific points**

1) L118 - they cannot track cloud development individually, there are a number of papers that use multiple polar-orbiting satellites to observe cloud development (e.g. Matsui et al., JGR, 2006; Meskhidze et al., ACP, 2009)

**Response: Thank you for your comment. We have rephrased these sentences and added**

**these references you mentioned in the manuscript.**

*"Tracking the development of convective cloud can provide more details of changes in cloud lifecycle. Attempts have been made to investigate differences in cloud properties between morning and afternoon using Terra and Aqua MODIS data (e.g. Meskhidze et al., 2009), but they cannot track the development of convective clouds beyond two snapshots per day. On the contrary, geostationary satellite data can provide images every 2.5-15 minutes, making it possible to track the evolution of convective clouds individually over the entire day"*

2) L189 - Is PM2.5 suitable for use as a CCN proxy? Presumably there has been some work done on this before, but I imagine the suitability as a proxy depends on aerosol type. Is that constant across this region, particularly between the lowlands and highlands?

**Response: It is an advantage that surface-measured PM$_{2.5}$ is available under all sky conditions over large areas, compared with optical properties from satellite-based passive remote sensors that can only be retrieved under cloud-free conditions, and the active sensors which have very limited coverage. That is the main reason we chose this dataset in this study.**

**It is true that aerosol type can affect the activation rate of aerosol particles, but the size distribution of aerosol particles is generally the dominant factor impacting activation under a specific supersaturation, based on Köhler theory. Besides, as eastern China is under continuous emissions of newly formed particles and precursor gases, particles tend to be coated with hydrophilic matter or grow into nucleation-mode easily, forming CCN and leading to more activations (Zhang et al. 2017). In addition, CCN concentrations are to some extent proportional to aerosol loading in many circumstances. Zhang et al. (2019) have found that the CCN number concentration increases with total aerosol number concentration and PM during haze events, so PM$_{2.5}$ concentration can also be partially representative of CCN concentration.**

**We have rephrased this part in section 2.2 as follows:**

*"On the other hand, particulate matter can be measured from the surface or aircraft under all-sky conditions and can provide dried particle masses which can minimize the influence of moisture on it.. Besides, as eastern China is under continuous emissions of newly formed particles and precursor gases, particles tend to grow into nucleation-mode easily, forming CCN and leading to activations. In addition, CCN concentrations are to some extent proportional to aerosol loading (Andreae, 2009; Liu and Li 2014). Zhang et al. (2019) found that the CCN number concentration increases with the total aerosol number concentration and PM during haze events. Particle size up to 10 μm may be much larger than the typical scale of CCN, so we consider particulate matter up to 1 μm (PM$_1$) or 2.5 μm (PM$_{2.5}$) in diameter are more appropriate to be taken as CCN proxies. Due to the limited availability of PM1 measurement in eastern China, we chose PM2.5 as an indicator of different CCN levels in the environment for this study."*

**We have aggregated the station-measured PM2.5 into grid boxes, thus the PM2.5 concentrations are different from one grid box to another. So, there are also differences between highlands and lowlands, and the higher terrains are always cleaner than lower places. Nevertheless, we are comparing the relatively high and low aerosol loading within each grid box, so convective clouds located within each grid box are classified regardless**

**of the terrain heights. More details are shown in response to comment 16).**

**References:**

*Andreae, M. O. (2009), Correlation between cloud condensation nuclei concentration and aerosol optical thickness in remote and polluted regions, Atmos. Chem. Phys., 9(2), 543-556, doi:10.5194/acp-9-543-2009.*

*Liu, J., and Z. Li (2014), Estimation of cloud condensation nuclei concentration from aerosol optical quantities: influential factors and uncertainties, Atmos. Chem. Phys., 14(1), 471-483, doi:10.5194/acp-14-471-2014.*

*Zhang, F., Wang, Y., Peng, J., Ren, J., Collins, D., Zhang, R., et al. (2017). Uncertainty in predicting CCN activity of aged and primary aerosols. Journal of Geophysical Research: Atmospheres, 122, 11,723–11,736. https://doi.org/10.1002/ 2017JD027058*

*Zhang, F., Ren, J., Fan, T., Chen, L., Xu, W., Sun, Y., et al. (2019). Significantly enhanced aerosol CCN activity and number concentrations by nucleation-initiated haze events: A case study in urban Beijing. Journal of Geophysical Research: Atmospheres, 10.1029/2019JD031457*

3) L206 - A number of meteorological variables are quoted here, but different ones are then described below. How do they relate to each other?

**Response: Some of the variables are used to calculate the ones described below. To reduce the ambiguity of expression, we have rewritten the introduction to the meteorological variables we use in section 2. Other changes to the meteorological variables are made, and the details can be seen in responses to comment 4) and 15).**

4) L215 - LTS has shown some skill at predicting stratiform cloud cover over the ocean (Klein and Hartmann, J. Clim, 1993). Is it really suitable for investigating convective clouds? It also has a dependence on the surface altitude, as the lapse rate is not usually the dry adiabatic lapse rate (e.g. Wood and Bretherton, J. Clim, 2006). Is it suitable for a study where the terrain height changes? L230 - Wang et al (2018) are also using other meteorological variables that include humidity information (such as CAPE). CAPE is not used here (but might be suitable for looking at convective clouds?)

**Response: By considering your comment, we have removed the LTS in this paper and change it into CAPE to represent the stability of atmosphere in different regions. Thus, we have replotted Figure 5, 13 and 14. And in Section 2, CAPE is introduced instead of LTS. We have added the following statement in Section 2:**

***"Convective Available Potential Energy (CAPE). CAPE is calculated from the ECMWF Integrated Forecasting System (IFS), by considering parcels of air departing at different model levels below the 350 hPa level (Hersbach et al., 2018). Numerous studies have shown that CAPE is a good indicator of atmospheric instability (e.g., Williams et al., 2002), and can reflect the potential ability to convective cloud formation and development."***

*Reference:*

*Hersbach, H., et al. (2018), ERA5 hourly data on single levels from 1979 to present, edited, Copernicus Climate Change Service (C3S) Climate Data Store (CDS),*

*doi:10.24381/cds.adbb2d47.*

*Williams, E., et al. (2002), Contrasting convective regimes over the Amazon: Implications for cloud electrification, Journal of Geophysical Research: Atmospheres, 107(D20), LBA 50-51-LBA 50-19, doi:10.1029/2001JD000380.*

5) L243 - Lee et al (2019) is looking at surface heterogeneity in a high-resolution model for a 28km domain. Is it a suitable reference relating the large-scale average vertical velocity to cloud-scale updrafts?

**Response: Per your suggestion, we have removed the reference here.**

6) L257 - This texture method looks like an interesting way to identify convective clouds, but I was not clear on what exactly is being done. Equations 3-6 are introduced, but apparently not then used or referred to. It is stated that the contrast at d=1 is used. Is this then the central-difference second derivative at each point? It would be good to have some way to visualize it for those not familiar with the GLCM.

**Response: Per your suggestion, we have removed the equations not used in this study and added a figure in the supplementary file to visualize the GLCM. The description is added in the figure caption of the GLCM.**

**We revised several parts of this paragraph to make the description of the calculations clearer:**

*"To define texture properties from the GLCM, several image statistical variables are derived from this matrix, including "contrast", "homogeneity", "energy", and "entropy", etc. (Haralick, 1979; Welch et al., 1988c; Baum et al., 1997; Bottino and Ceballos, 2014). The "contrast" measures the intensity contrast between a pixel and its neighbors, assessed over the entire image. "Homogeneity" measures the closeness to the diagonal of the GLCM element distribution. "Energy", also termed the angular second moment, measures the complexity of the image, and "entropy" measures the degree of randomness, evaluated over the entire image.*

*The objective of this section is to identify new and mature convective clouds. As the edges of convective clouds tend to be very sharp (Purdom, 1976) in visible channel images, large differences between i and j occur. Thus, large 'contrast' values can be found (Equation 2) at the edge of convective clouds. The formula is:*

$$Contrast = \sum_{i,j} |i-j|^2 GLCM(i,j) \qquad (2)$$

*We use the mean "contrast" data at d=1 in the four directions (θ=0°, 45°, 90°, and 135°) calculated from the visible channel (0.64 μm) to identify convective clouds."*

7) L300 - Which wavelength is used to calculate the contrast?

**Response: the 0.64μm visible channel is used to calculate the contrast, and we have added an illustration in this line.**

*"We use the mean "contrast" data at d=1 in the four directions (θ=0°, 45°, 90°, and 135°) calculated from the visible channel (0.64μm) to identify convective clouds."*

8) L302 - It was stated earlier that the texture method could reduce the need for thresholds,

but some are still included. How much extra does the texture filter add beyond these filters? L326 suggests that the majority of the difference from MOD35 comes from the cirrus filtering, as the TCT-CID method selects the convective cores.

**Response: The method is designed by combining texture analysis and thresholding to identify both small and deep convective clouds. The texture analysis helps find shallow convective clouds, such as stratocumulus and convective initials. And the thresholds are mostly used in selecting clusters that are more likely to be convective clouds and in identifying flat cloud tops that have low contrast values and cannot be identified by the texture analysis. We applied a split-window method to exclude cirrus cloud because some of the high clouds also have sharp edges and are bright in VIS images. The texture analysis and thresholds can also include these high clouds, but we cannot take them into account when identifying convective clouds. Although the addition of this filtering will exclude the anvils of some mature convective clouds, our method still captures the main body of convective clouds. We have also added a discussion as follows:**

*"The method is designed to find sharp edges and bright clusters in VIS images firstly, but as some of the high clouds may also fit these conditions, which are not the subject of this study, we added a split-window filter to exclude this kind of cloud. However, the addition of this filter will also exclude the anvils of some mature convective clouds and some of the shallow clouds, so the cloud area identified by our method is smaller than the cloudy area found by the MODIS cloud mask and the capability of shallow convection identification is limited. Nevertheless, the majority of convective clouds are well captured by our method."*

**In addition, we performed a case comparison between our identification method with two threshold-only methods in Figure R2. The convective cloud mask of our method is shown in red in Figure R2b, R2c and R2d, whereas two threshold-only method is shown as blue and cyan in Figure R2c and R2d, respectively. The thresholds in Figure R2c are the same to our method (with area smaller than 10,000 pixels, mean VIS reflectance larger than 0.75, maximum VIS reflectance larger than 0.9, and brightness temperature difference between 11.2 μm and 12.4 μm channel higher than -4k), the only difference is that we turned off the clustering process involving texture properties. Another threshold-only method is to identify convective cloud with a 11.2 μm brightness temperature threshold, which is widely used in identifying mesoscale convective systems. As previous studies have taken thresholds ranging from 208-258K (e.g. Maddox et al. 1980; Hodges & Thorncroft, 1997; Machado et al., 1998; Liang et al. 2008; Ai et al., 2016; Chen et al. 2019), we chose the mostly used 235 K here (Figure R2d). We can find that our method can identify both the mature convective clouds and the shallow convections, which is better than only using either VIS or brightness temperature threshold.**

[Figure]

**Figure R2. Comparison between the convective cloud (CC) mask identified by TCT-CID and two threshold-only methods, at 13:40 LT on July 30th, 2016. (a) The true color image, (b) CC mask by TCT-CID method (shown in red), and comparison between this CC mask (red) with a method of (c) threshold same to TCT-CID but with the clustering process turned off (blue), and (d) threshold of brightness temperature < 235K (cyan).**

**References:**

*Maddox, R. A. (1980), Meoscale Convective Complexes, Bulletin of the American Meteorological Society, 61(11), 1374-1387, doi:10.1175/1520-0477(1980)061<1374:mcc>2.0.co;2.*

*Hodges, K. and Thorncroft, C. (1997). Distribution and statistics of the African mesoscale convective systems based on the ISCCP Meteosat Imagery. Monthly Weather Review, 125, 2821–2837. https://doi.org/10.1175/1520-0493(1997)125 <2821:DASOAM>2.0. CO;2*

*Machado, L. A. T., Rossow, W. B., Guedes, R. L., & Walker, A. W. (1998). Life cycle variations of mesoscale convective systems over the Americas. Monthly Weather Review, 126, 1630–1654. https://doi.org/10.1175/15200493(1998)126<1630:LCVOMC>2.0.CO;2*

*Laing, A. G., Carbone, R., Levizzani, V., & Tuttle, J. (2008). The propagation and diurnal cycles of deep convection in northern tropical Africa. Quarterly Journal of the Royal Meteorological Society, 134, 93–109. https://doi.org/10.1002/qj.194*

*Ai, Y., W. Li, Z. Meng, and J. Li (2016), Life Cycle Characteristics of MCSs in Middle East China Tracked by Geostationary Satellite and Precipitation Estimates, Monthly Weather Review, 144(7), 2517-2530, doi:10.1175/mwr-d-15-0197.1.*

*Chen, D., et al. (2019), Mesoscale Convective Systems in the Asian Monsoon Region From Advanced Himawari Imager: Algorithms and Preliminary Results, Journal of Geophysical Research: Atmospheres, 124(4), 2210-2234, doi:10.1029/2018jd029707.*

9) L311 - "Numerically smaller" is a bit ambiguous. Do you mean "less than"?

**Response: We have changed the phrase "numerically smaller" into "less than".**

10) L343 - Does this mean that the TCT-CID method has an error that depends on solar zenith angle? That might be an issue for looking at the diurnal cycle. Does it have a large impact?

**Response: We would admit that the VIS reflectance depends on the solar zenith angle. However, as we applied texture analysis in our method, the error may be reduced. From Figure S2, we can see that the method shows robust identification results from 08:00-17:00, especially for deep convective clouds. Although other cloud types can be included in the identification results, the deep convective clouds are still the category that most closely matches ISCCP products after 16:00. So, the changes in solar zenith angle may have an impact on the results, but may not be significant.**

11) L366 - Yang and Slingo, M. Weather Rev. (2001) might also be a useful reference, although it only covers the southern part of China

**Response: Thank you for your suggestion, we have added the reference in the manuscript.**

12) L371 - You probably only need to cite one edition of "Cloud Dynamics"

**Response: Per your suggestion, we have removed the 1993 edition.**

13) L373 - The gray lines were a bit difficult to see on my screen. Perhaps you could make them black and remove the provinces from the map?

**Response: We have replotted Figure 4 to change the color bar and removed the provinces. The contour of 1000m terrain height is shown in black instead.**

14) L388 - The LTS has an in-built dependence on the above-inversion lapse rate and hence the altitude of the 700hPa level. This could explain part of the dependence on the altitude.

**Response: We have changed the variable LTS into CAPE in this study, please see response to comment 4).**

15) L396 - The split of RH and updraft levels for the high and low populations might generate an artificial difference between them. Might it be more appropriate to use an updraft and RH at a specific height above the surface instead?
L399 - Related to L396 - the difference may be that different variables are being compared? This makes the following interpretation difficult.

**Response: Thanks for pointing out the problem in the methodology used in this study. We have attempted to address the concerns raised by you as follows:**

We have changed RH at 850hPa and 700hPa and updraft at 925hPa and 850hPa levels into RH and ω at 800hPa and 900hPa for regions with terrain height >1000m and <1000m instead, so that we can roughly get the RH and ω fields about 1000m above the surface. So, we have replotted Figure 5 and Figure 13, 14, and we have also rewritten the descriptions for these figures as follows:

*Line 250-252: "We use ω800 and ω900 to represent the low-level dynamical conditions for terrain above and below 1000 m, respectively, so that we can roughly assess the ω about 1000m above the surface in these regions despite varying terrain height."*

16) L427 - Is there a regional bias in the clean/polluted pixels? Does this end up comparing N. China to S. China (for example)?

**Response: It is true that some areas are always cleaner or more polluted than others. But in this study, we have found the upper and lower quarter of PM2.5 values for each site and aggregated them into 0.4°×0.4° gridded fields to act as clean and polluted threshold fields, so that each pixel has a specific clean and polluted threshold. All the convective clouds with their centroids located in the grid box are classified using the clean and polluted threshold value for that grid box. So, we are comparing the relative polluted and clean cases within the box, and we do not end up comparing one place to another. To make this point clearer, we have rephrased the statements in the first paragraph of Section 4.2 as follows:**

*"We then aggregate them into a 0.4°×0.4° grid using nearest neighbor interpolation to create clean and polluted fields. The average clean and polluted thresholds over the ROI are shown as dashed and solid red lines in Figure 6. All the convective clouds with their centroids located in the grid box are classified using the clean and polluted threshold value for that grid box. The centroids of convective clouds located within a grid box having $PM_{2.5}$ concentration greater (less) than the polluted (clean) threshold are deemed as polluted (clean) convective clouds."*

17) L451 - It is interesting that the change is first observed in smaller clouds and this is a useful result. I am not clear how aerosol-radiation interactions are responsible though.

**Response: We have removed this statement here and add inferences of this phenomenon for Figure S4:**

*"This may also explain the phenomenon in Figure S4. As solar radiation begins to wane in the afternoon, surface heating weakens, the high aerosol loading would suppress convection via the dominance of radiative effect, thus inhibit the formation of small convective clouds first."*

18) L467 - An extra explanation step would be useful to help the reader. How does a decrease in surface temperature lead to a suppression of the vertical moisture flux?

**Response: We have rewritten the sentence to make the description of the processes clearer and we also added references to support these statements:**

*"However, when aerosol concentration is higher, the attenuation of solar radiation by aerosol particles decreases the surface temperature, and atmospheric heating decreases the lapse rate, thus inhibiting sensible heat flux (Feingold et al., 2005). The decrease in the*

*surface temperature also reduces evaporation (Koren et al., 2004), so that the atmosphere becomes drier and more stable, which suppresses convection."*

*References:*

*Feingold, G., et al. (2005). "On smoke suppression of clouds in Amazonia." Geophysical Research Letters 32(2).*

*Koren, I., et al. (2004). "Measurement of the effect of Amazon smoke on inhibition of cloud formation." Science 303(5662): 1342-1345.*

19) L500 - This would be good to check.

**Response: We have found the mistake in this sentence and have rewritten it into:**

*"But as both the surface and the air above are generally moister before noon time (Figure 5q-z2), the higher moisture may suppress cloud droplet evaporation and thus keep CCF from sharply decreasing."*

20) L510 - I am not sure what the semicolon means here (rather than a comma)

**Response: We have changed the equations, please see response to comment 22).**

21) L514 - This is a little bit ambiguous, is the daytime 0800-1700, or is it defined using solar zenith angle?

**Response: We mean the time period of 08:00-17:00 here, and have changed the phrase "during daytime" into "during the 08:00-17:00 daytime period".**

22) L526 - How are the values in Fig. 9 calculated? My reading of Eq. 8 would make it constant. In Fig. 9, the values presented are averaged over the whole region, so NC_CF / _i_jNCCF = _i_jNc(P)(i;j;h;t) _i_jNc(P)(i;j;h;t) = const.

Clearly I am missing something here, as this would lead to the NCCF being constant in time and altitude.

**Response: Thanks for pointing out the problem in the equations. We have rewritten equation 7 and 8 to make the $CCF_{c(p)}$ and $NCCF_{c(p)}$ consistent with the values in Figure 9 and 10, as originally intended. And rewritten the descriptions of equations 3 and 4:**

*"The CCFs at different TH in both polluted and clean conditions are shown in Figure 9. The CCF under clean (or polluted) conditions within each elevation bin h is calculated using the formula shown below:*

$$CCF_{C(P)}(h,t) = average\left(\frac{N_{C(P)}|_h(i,j,t)}{N_{total}|_h(i,j)}\right) \times 100\% \qquad (3)$$

*where $N_{C(P)}|_h$ (i,j,t) represents the number of convective clouds occurring under clean (C) or polluted (P) conditions in the (i,j)th pixel box in the ROI during hour t, and $N_{total}|_h$(i,j) represents the total number of convective clouds observed in each pixel box during the 08:00-17:00 daytime period. For each CCF bin in Figure 9, i and j denote the pixels within the region having elevation h."*

**And for equation 8 we have rewritten as:**

*"The normalized CCF (NCCF) is computed as:*

$$NCCF_{C(P)}(h, t) = average\left(\frac{N_{C(P)}|_h(i,j,t)}{\sum_m \sum_n N_{C(P)}(m,n,t)}\right) \times 100\% \qquad (4).$$

*The NCCF is also calculated with i and j at each elevation bin h. Unlike CCF (Equation 3), the denominator for NCCF (Equation 4) is not summed over all studied times-of-day but over the entire ROI (the size of which is m×n pixels) within each hour."*

23) L564 - How are clean conditions defined here? They were previously defined at a gridbox level, but this is apparently an average over the whole regions?

**Response: The clean and polluted conditions are defined in *Lines 440-447*, we have horizontally classified each reanalysis data grid cell into clean and polluted cases using the clean and polluted threshold fields calculated from the surface PM$_{2.5}$ concentrations in section 4.2. We then taken the zonal and meridional average of all the polluted or clean cases to plot Figure 12. To make this point clearer, we have rewritten the descriptions of Figure 12 as follows:**

*"We have classified each reanalysis data grid cell into clean and polluted cases using the clean and polluted threshold fields described in section 4.2. The meridional-vertical mean distribution of relative humidity and wind are then calculated (Figure 12)."*

24) L568 - This is a good example of the correlation vs causation issue. While the reduced above cloud RH in polluted conditions could lead to a "restraint of convective cloud development due to the inhibition of (the) aerosol microphysical effect", a simpler explanation is that the reduced above cloud RH itself limits convection, which doesn't require an aerosol impact on cloud. It is not clear which explanation is correct (probably a little of both), but it is not clear to me that aerosols are responsible for the aerosol-CCF relationship.

**Response: Thanks for this suggestion, we have rephrased the sentence as follows:**

*"But under polluted conditions, the southerly wind is weaker, and the relative humidity above this region is lower, which may provide a restraint on convective cloud development. Both the drier conditions and the increase in aerosol loading are likely to contribute to the inhibition of convective cloud."*

25) L609 - "may prove", "are probably". I am not sure that the evidence is there to make this strong a claim. There is still a considerable correlation between the aerosol and meteorological environment (Fig. 13)

**Response: We have changed the words "may prove" and "are probably" into "perhaps indicate" and "are potentially" in this section.**

26) L619 - Updraft is also related to the activation - if this is an aerosol effect, an increased updraft can lead to a larger sensitivity to aerosol (moving to an aerosol-limited environment).

**Response: We have rephrased the sentence per your suggestion:**
*"This pattern is likely to indicate that more aerosol particles can be activated or entrained into the clouds from the boundary layer when uplift is stronger, which in turn might strengthen the aerosol microphysical effect."*

27) L642 - "testing whether the results are mainly due to aerosol effects" is the same things as establishing causality (as I read it). I would perhaps consider - "However, establishing a correlation between aerosol and cloud properties is only a first step;"

**Response: We have rewritten the sentence:**

*"However, identifying the apparent correlations between convective cloud and aerosol loading is only a first step, supporting the hypothesized mechanisms by which aerosol affects convective cloud occurrence. The mechanism by which the aerosol and meteorology interact is another important question, that is beyond the scope of the current study."*

28) L690 - Perhaps the pattern is "...consistent with the combined action..."? Given the correlations between aerosol and meteorology (Fig. 13), it is not clear that these correlations imply causality. This follows in later parts of the conclusions. Correlations have been shown (and these are very interesting!), but it is not yet clear these are the result of an aerosol impact on clouds.

**Response: Per your suggestion, we have rephrased the sentence as follows:**

*"We find that the meteorological variations driven by diurnal solar radiation changes and topography are among the potential reasons for changes in the relationship between convective cloud and aerosol loading, which could be related to the changes in the relative strength of aerosol microphysical and radiative effects."*

**3 Additional references**

Boucher, O., and J. Quaas (2012), Water vapour affects both rain and aerosol optical depth, Nat. Geosci., 6(1), 4–5, doi:10.1038/ngeo1692.

Gryspeerdt, E., J. Quaas, T. Goren, D. Klocke, and M. Brueck (2018), An automated cirrus classification, Atmos. Chem. Phys., 18(9), 6157–6169, doi:10.5194/acp-18-6157-2018.

Joos, H., P. Spichtinger, U. Lohmann, J.-F. Gayet, and A. Minikin (2008), Orographic cirrus in the global climate model ECHAM5, J. Geophys. Res., 113, D18205, doi:10.1029/2007JD009605.

Koren, I., Y. J. Kaufman, D. Rosenfeld, L. A. Remer, and Y. Rudich (2005), Aerosol invigoration and restructuring of Atlantic convective clouds, Geophys. Res. Lett., 32, 14828, doi:10.1029/2005GL023187.

Koren, I., G. Feingold, and L. A. Remer (2010), The invigoration of deep convective clouds over the Atlantic: aerosol effect, meteorology or retrieval artifact?, Atmos. Chem. Phys., 10(18), 8855–8872, doi:10.5194/acp-10-8855-2010.

Meskhidze, N., L. A. Remer, S. Platnick, R. Negrón Juárez, A. M. Lichtenberger, and A. R. Aiyyer (2009), Exploring the differences in cloud properties observed by the Terra and Aqua MODIS Sensors, Atmos. Chem. Phys., 9(10), 3461–3475, doi:10.5194/acp-9-3461-2009.

Williams, E. et al. (2002), Contrasting convective regimes over the Amazon: Implications for cloud electrification, J. Geophys. Res., 107(D20), 8082, doi:10.1029/2001JD000380.

Wood, R., and C. S. Bretherton (2006), On the Relationship between Stratiform Low Cloud Cover and Lower-Tropospheric Stability, J. Climate, 19(24), 6425–6432, doi:10.1175/JCLI3988.1.

Yang, G.-Y., and J. Slingo (2001), The Diurnal Cycle in the Tropics, M. Weather Rev., 129,

784, doi:10.1175/1520-0493(2001)129<0784:TDCITT>2.0.CO;2.

---

## Author Response (AR2)

**Responses to Comments**

1) Please move the presentation and description of Eqs. 3 and 4 into the methodology section.
**Response: We have moved the equations into the methodology section, and to distinguish the CCFs described in these equations with the CCFs in section 4.1 and 4.2, we have added corner marks to these variables in the context as:**

*"3.3 Convective cloud fraction at different terrain heights in polluted or clean environment*

*In order to isolate the potential effect of topography on the aerosol-convective-cloud relationship, we investigate in this study the convective cloud fraction (CCF) changes along with TH at different levels of aerosol loading. The CCF under clean (or polluted) conditions (CCFC(P)) within each elevation bin h is calculated using the formula shown below:*

$$CCF_{C(P)}(h,t) = average\left(\frac{N_{C(P)}|_h(i,j,t)}{N_{total}|_h(i,j)}\right) \times 100\% \tag{3}$$

*where NC(P)|h(i,j,t) represents the number of convective clouds occurring under clean (C) or polluted (P) conditions in the $(i,j)^{th}$ pixel box in the ROI during hour t, and $N_{total}|_h$ (i,j) represents the total number of convective clouds observed in each pixel box during the 08:00-17:00 daytime period. For each CCF bin, i and j denote the pixels within the region having elevation h. Sample sizes are shown in Figure S3.*

*By normalizing the occurrence frequencies by the total number of polluted and clean cases within each hour, respectively, we explore how topography changes the polluted and clean convective clouds spatially. The normalized $CCF_{C(P)}$ ($NCCF_{C(P)}$) is computed as:*

$$NCCF_{C(P)}(h,t) = average\left(\frac{N_{C(P)}|_h(i,j,t)}{\sum_m \sum_n N_{C(P)}(m,n,t)}\right) \times 100\% \tag{4}$$

*The $NCCF_{C(P)}$ is also calculated with i and j at each elevation bin h. Unlike $CCF_{C(P)}$ (Equation 3), the denominator for $NCCF_{C(P)}$ (Equation 4) is not summed over all studied times-of-day but over the entire ROI (the size of which is m×n pixels) within each hour. NCCF focuses more specifically on how CCF at a given location and terrain elevation compares with all locations at the same elevation and the same time, reducing the influence of diurnal variation and emphasizing elevation-related differences. As such, the difference in $NCCF_{C(P)}$ between clean and polluted cases reflects the difference caused by topography when the overall environment is under clean or polluted conditions."*

**And in section 4.3 we have also rewrote the discussions as follows:**

*"The CCFs at different TH in both polluted ($CCF_P$) and clean ($CCF_C$) conditions are shown in Figure 9 (calculated by Equation 3). We find that the CCF difference between polluted and clean conditions generally agrees with Figure 7 in that $CCF_P$ is higher in the morning, lower in the afternoon, and the differences are statistically greatest in early morning and late afternoon. In addition, the differences between $CCF_P$ and $CCF_C$ vary considerably along with increasing TH, which may indicate that the effects of topography and air quality on CCF co-vary, and the impact of topography might be much stronger compared with increased aerosol loading.*

*There is also another aspect of these phenomena. Because the elevation-related response reverses over the day (e.g., Figure 9), we applied $NCCF_{C(P)}$ to reduce the impact of diurnal*

*variations. **We can see from Figure 10 that below the elevation of 500 m, most of the convective clouds are suppressed under polluted conditions, whereas over regions with terrain height greater than 1000 m, the amount of convective cloud under polluted conditions is significantly larger before 14:00 LT. This phenomenon may partly explain the results shown in Figure 7, where complex topography plays an important role in the aerosol effect on convective clouds. Under polluted conditions, convective clouds over lower terrain are much easier to suppress, whereas over elevated terrain, convective clouds are more likely to be invigorated."***

2) Please double-check the caption of Figure 6: Do the red lines and numbers really refer to the mean values of the bottom and top quarter or rather to the threshold values to separate them from the 25% to 75% range?

*Response: We have checked the caption. The red lines and numbers in Figure 6 are the mean values of the bottom and top quarter, and their values are just close to the 75% and 25% of the distribution in Figure 6.*